# Peroxisomal core structures segregate diverse metabolic pathways

Nils Bäcker[1,2,12], Julia Ast[2,12], Domenica Martorana[2,12], Christian Renicke[2,3], Jil Berger[1,4], Cristopher-Nils Mais[1,4], Marvin Christ[1,4], Thorsten Stehlik[2], Thomas Heimerl[1,4], Valentin Wernet[5], Christof Taxis[1,6], Jan Pané-Farré[1,4], Michael Bölker[1,2], Judith M. Klatt[1,4,7], Björn Sandrock[2], Kay Oliver Schink[8,9,10] ✉, Gert Bange[1,4,11] ✉ & Johannes Freitag[1,2] ✉

Peroxisomes are single membrane-bounded oxidative organelles with various metabolic functions including β-oxidation of fatty acids. Peroxisomes of many species confine certain metabolic enzymes into sub-compartments sometimes visible as electron dense cores. Why these structures form is largely unknown. Here, we report that in the smut fungus *Ustilago maydis* detergent resistant core structures are enriched for different enzymes excluding several key enzymes of the β-oxidation pathway. This confinement contributes to generation of peroxisome subpopulations that differ in their enzyme content. We identify short amino acid motifs necessary and sufficient for protein self-assembly into aggregates in vitro. The motifs trigger enrichment in cores in vivo and are active in mammalian cells. Perturbation of core assembly via variation of such motifs affects peroxisome function in *U. maydis* strains challenged with fatty acids. Thus, protein core structures serve to compartmentalize the lumen of peroxisomes thereby preventing interference of biochemical reactions. Metabolic compartmentalization of peroxisomes via assembly of specific proteins may occur in other organisms as well.

Peroxisomes are organelles that provide a safe location for oxidative reactions such as breakdown of fatty acids but are also involved in versatile other biochemical pathways often specific for particular organisms[1–3]. In humans, peroxisomes are essential and mutation of biogenesis proteins (Peroxins) causes disorders such as the Zellweger syndrome[4–6].

Protein import into the peroxisomal lumen requires either of two cytosolic targeting factors, Pex5 or Pex7[7,8]. Both recognizes specific sequence motifs either located at the Carboxy (*C*)-terminal end of the cargo protein (Pex5: peroxisomal targeting signal type 1 (PTS1)) or within the Amino (*N*)-terminal part (Pex7: peroxisomal targeting signal type 2 (PTS2))[9,10]. The prototype PTS1 is a tripeptide with the amino acid sequence Ser-Lys-Leu[11]. PTS1 proteins translocate in a folded and even oligomeric state through a channel that may resemble the interior of the nuclear pore, forms a hydrogel in vitro and involves phase separation of key components[12–16].

[1]Center for Synthetic Microbiology (SYNMIKRO), Philipps-University Marburg, Marburg, Germany. [2]Department of Biology, Philipps-University Marburg, Marburg, Germany. [3]Department of Genetics, Stanford University School of Medicine, Stanford, CA, USA. [4]Department of Chemistry, Philipps-University Marburg, Marburg, Germany. [5]Department of Microbiology, Institute for Applied Biosciences, KIT, Karlsruhe, Germany. [6]Health and Medical University Erfurt, Erfurt, Germany. [7]Microcosm Earth Center, Philipps-University Marburg & Max-Planck-Institute for terrestrial Microbiology, Marburg, Germany. [8]Centre for Cancer Cell Reprogramming, Faculty of Medicine, University of Oslo, Montebello, Oslo, Norway. [9]Department of Molecular Cell Biology, Institute for Cancer Research, Oslo University Hospital, Montebello, Oslo, Norway. [10]Department of Molecular Medicine, Institute of Basic Medical Sciences, Faculty of Medicine, University of Oslo, Oslo, Norway. [11]Max-Planck-Institute for terrestrial Microbiology, Molecular Physiology of Microbes, Marburg, Germany. [12]These authors contributed equally: Nils Bäcker, Julia Ast, Domenica Martorana. ✉e-mail: k.o.schink@medisin.uio.no; gert.bange@synmikro.uni-marburg.de; johannes.freitag@biologie.uni-marburg.de

Fungal peroxisomes comprise diverse specialized metabolic pathways, e.g., for secondary metabolite and siderophore biosynthesis, methanol degradation and carbohydrate metabolism[2,17–20]. A peculiar subpopulation of peroxisomes identified in *Neurospora crassa* are Woronin bodies, which act as a plug to seal septal pores in wounded fungal hyphae[21,22]. Differentiation of peroxisomes into Woronin bodies involves self-assembly of the Hex1 protein into large crystals[23,24]. Peroxisome subpopulations are also found in other organisms. For instance, characterization of isolated rat liver peroxisomes revealed peroxisomal populations with different protein content[25]. In yeast, it was shown that the peroxisome population is heterogenous and that their content is changing with age[26]. In plants, different types of peroxisomes can emerge during development[27,28].

Another feature of peroxisomes is internal compartmentalization. Defined subdomains consisting of peroxisomal membrane proteins have been observed[29]. Additionally, early electron microscopy and biochemical studies demonstrated that urate oxidase and further oxidative enzymes are major constituents of electron-dense and detergent resistant core structures[30–35]. Similar proteinaceous cores enriched for alcohol oxidases were identified in methylotrophic yeast[36]. However, why peroxisomes form such structures is unknown.

Here, we show that several enzymes enrich in detergent-resistant core structures inside of peroxisomes thereby enabling formation of intraluminal subdomains and eventually subpopulations of peroxisomes. We could identify amino acid motifs required for aggregation in vitro and core formation in vivo indicating that subdomains form by self-assembly. We found that this type of internal compartmentalization is required for efficient metabolism of oleic acid.

## Results

### Formation of peroxisomal subpopulations and luminal subdomains

We have previously shown that in *Ustilago maydis* two acyltransferases, Mac1 and Mac3, involved in biosynthesis of surface-active mannosylerythritol lipids (MELs), are localized to peroxisomes[37–40]. Examination of cells expressing green fluorescent GFP-Mac1 and GFP-Mac3 revealed that both enzymes are not equally distributed among all peroxisomes that contain the peroxisomal marker protein mCherry-SKL but are highly enriched only in a subset of organelles (Fig. 1a). To get insight into this phenomenon we first studied the behavior of subpopulations during peroxisome proliferation. Growth on the C18 fatty acid oleic acid initiates peroxisome proliferation in many fungi[41,42]. Indeed, incubation of cells in oleic acid medium substantially reduced the colocalization of Mac proteins with mCherry-SKL (Fig. 1b). Analysis of fluorescent microscopic images revealed only low correlation for the red and the green fluorescent signals (Fig. 1c and Fig. S1). Many of the mCherry-SKL containing peroxisomes were even lacking either of the Mac proteins (Fig. 1b). Reciprocal exchange of fluorescent tags did not alter our results indicating that the different peroxisome populations are not caused by properties of the fluorescent proteins used (Fig. S2a). Moreover, we co-expressed GFP-Mac3 and mCherry-Mac3 or GFP-SKL and mCherry-SKL and observed a high degree of colocalization for each of the pairs (Fig. S2b–d). Previously we have shown that GFP-Mac1 under control of its own promoter is fully functional and only expressed under nitrogen starvation[39]. We confirmed accumulation of GFP-Mac1 in a peroxisome subpopulation also via this construct upon growth in medium lacking nitrogen (Fig. 1d). This observation indicates that distinct populations of peroxisomes form in *U. maydis*, providing evidence for a naturally occurring biological phenomenon.

GFP-Mac1 and GFP-Mac3 were enriched in foci that colocalized only partially with the peroxisomal lumen stained by mCherry-SKL (e.g., Fig. 1b). Thus, we analyzed suborganellar localization of GFP-Mac1 and GFP-Mac3 by super-resolution imaging using structured illumination microscopy (SIM)[43]. Analysis of SIM images revealed that

both fusion proteins are concentrated in subdomains often located at the periphery of peroxisomes, while mCherry-SKL and GFP-SKL were localized in the entire organellar lumen (Fig. 1e). A comparable localization was also observed when the peroxisomal membrane protein Pex12-GFP was used as a marker (Fig. S2e, f). We visualized the formation of subdomains by 3D-reconstructions (Fig. 1f and Fig. S2g). Foci containing Mac1/3 were stable over time (Fig. S2h and Movie 1). Thus, Mac1 and Mac3, which localize to peroxisome subpopulations, and enrich in stable peroxisomal subdomains.

### A short peptide motif in Mac3 necessary and sufficient for formation of peroxisomal subdomains after import

Next, to test whether peroxisome subpopulations emerge from selective import into only a subset of peroxisomes, or from processes that occur within the peroxisomal lumen, we performed a pulse chase experiment. While mCherry-SKL was constitutively expressed, GFP-Mac3 was expressed under control of the arabinose-inducible and glucose-repressible promoter $P_{crg}$[44] to follow the localization of newly synthesized protein over time. Furthermore, we created a GFP tagged version of acyl-CoA oxidase with $P_{crg}$ (Aox1; UMAG_02208). Upon constitutive expression GFP-Aox1 is evenly distributed among peroxisomes and, thus represents a suitable control (Fig. S3a). Expression of GFP fusion proteins was induced through incubation in arabinose-containing medium (Fig. 2a). By shifting cells to glucose conditions gene expression was repressed, and the localization of fluorescent proteins was analyzed by SIM and by epifluorescence microscopy at several time points (Fig. 2a, b and Fig. S3b). Both GFP-Mac3 and GFP-Aox1 were first uniformly distributed among peroxisomes (Fig. 2a). At later time points, GFP-Mac3, but not GFP-Aox1, accumulated in distinct foci, and colocalization was reduced (Fig. 2a, b). We noticed that GFP-Mac3 seems to be less stable over time compared to GFP-Aox1. We, therefore, repeated the chase experiment and followed the abundance of GFP-Mac3 and GFP-Aox1 by immuno-blotting confirming reduced stability (Fig. 2c). Hence, selective luminal degradation may contribute to the formation of subpopulations. It was established previously that peroxisomes contain a protease of the LON family that degrades misfolded peroxisomal proteins[45]. Deletion of the ortholog from the genome of *U. maydis* did not affect formation of subpopulations (Fig. S4a, b). So far our data suggest that subdomains form after import and that formation of peroxisome subpopulation coincides with degradation of GFP-Mac3 and segregation of subdomains upon peroxisome division (Fig. 1a, b). This interpretation is substantiated by our data that only one third of the peroxisomes contain Mac proteins and usually one subdomain per organelle (Fig. S4c).

In addition, we tested the impact of the PTS1 import pathway via a complementary approach. The C-terminal 12 amino acids (dodecamers) of PTS1 proteins contribute to the efficiency of the targeting signal[46]. To examine if PTS1 import contributes to formation of subpopulations we analyzed the localization of GFP fused to the C-terminal PTS1 containing dodecamers of Mac1 and Mac3, respectively (GFP-PTS1$_{Mac1}$ and GFP-PTS1$_{Mac3}$) (Fig. 2d). We found that GFP-PTS1$_{Mac3}$ accumulated in subpopulations and was concentrated in subdomains. GFP-PTS1$_{Mac1}$ was homogenously distributed inside of the lumen (Fig. 2d). Hence, only the C-terminal dodecamer of Mac3 includes information sufficient for focal enrichment. In Mac1 such a signal could be embedded elsewhere in the protein.

Truncation analysis revealed that addition of the C-terminal nonameric peptide of Mac3 significantly enhanced colocalization of GFP with mCherry-SKL compared to the dodecameric peptide (Fig. S5a, b). A short amino acid motif inside of the Mac3 derived dodecamer−consisting of the amino acid stretch Thr-Ile-Ile-Val (TIIV)− was even sufficient for efficient focal accumulation of GFP in peroxisome subdomains and subpopulations (Fig. 2e and Fig. S5c). Insertion of this sequence motif between GFP and a canonical SKL containing PTS1 motif was enough for formation of foci (Fig. 2e and Fig. S5c).

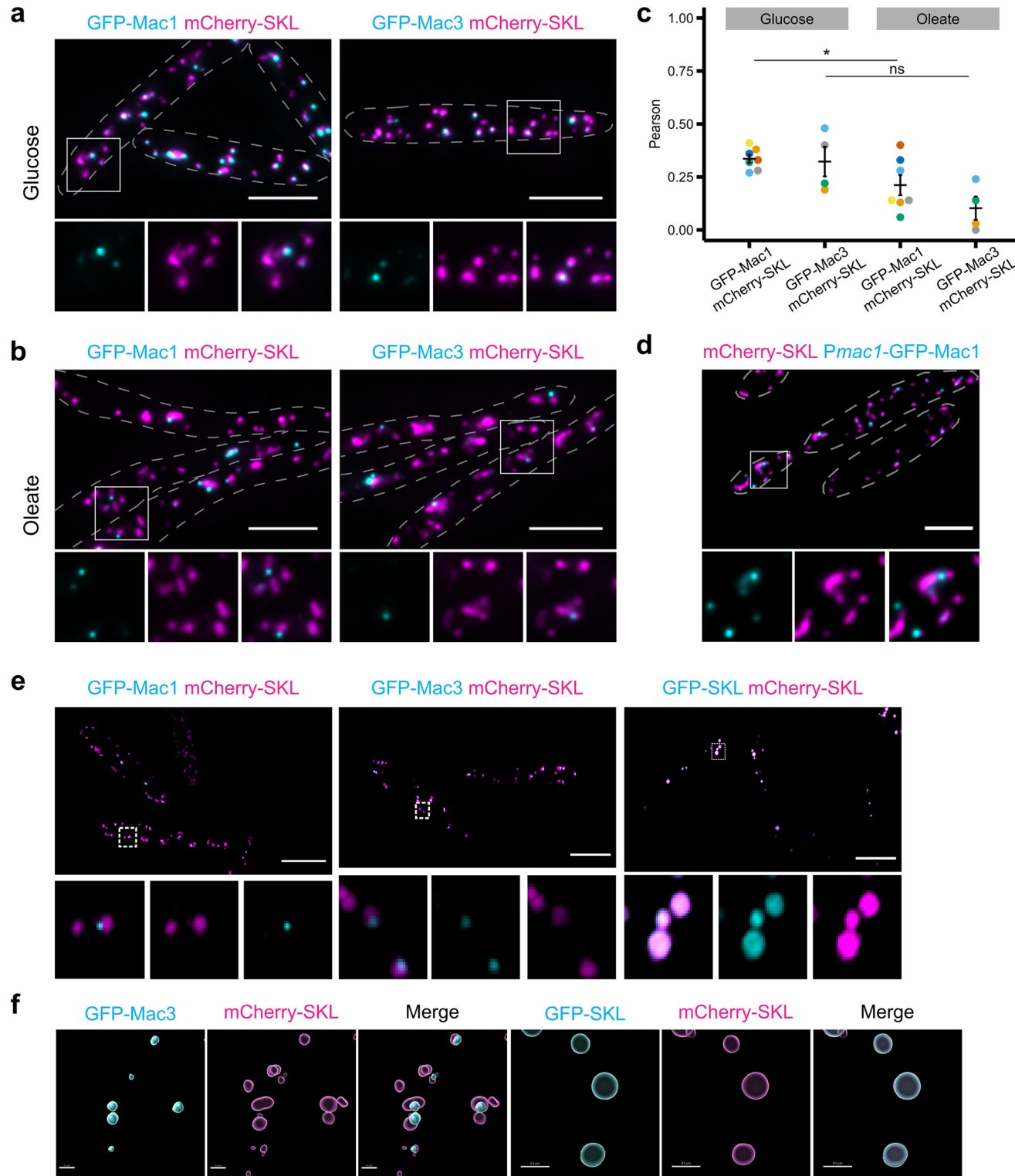

Fusion of the TIIV-tetrapeptide to the C-terminus of a PTS2-GFP reporter protein[47] resulted in a similar pattern of localization (Fig. 2f and Fig. S5e). Even fusion of a mitochondrial marker protein to TIIV triggered enrichment of GFP in focal structures in or at mitochondria (Fig. S5d) suggesting that functionality of this element is not restricted to a specific cellular location. Expression of GFP-Mac1 and GFP-Mac3 in mutants that fail to import PTS1 cargo ($\Delta pex5b$)[47] lead to predominantly cytosolic localization of both fusion proteins (Fig. S5f). Foci were only detected occasionally (Fig. S5f). Thus, local

concentration via peroxisomal import or mitochondrial import is likely a trigger for accumulation in foci.

To determine the critical residues in the TIIV motif we constructed variants with single amino acid substitutions. Exchange of the first and last residue with alanine resulted in enhanced colocalization with mCherry-SKL. Substitution of the last residue with arginine had a stronger effect (Fig. S6a and S6b). Therefore, a full-length version of GFP-Mac3 containing the V to R substitution (Mac3-$V_{570}$R) was constructed and a high degree of colocalization with mCherry-SKL was

**Fig. 1 | Acyltransferases Mac1 and Mac3 enrich in peroxisome subpopulations and peroxisomal subdomains.** *U. maydis* strains expressing N-terminally GFP-tagged versions of Mac1 and Mac3 (cyan) and the peroxisomal marker protein mCherry-SKL (magenta) were inspected by epifluorescence microscopy. Representative images of cells grown in glucose (**a**) or oleate (**b**). Full images are shown as overlays of two channels. For insets single channels and merged channels are provided. Scale bars: 5 μm. **c** Quantifications show Pearson's correlation coefficients of GFP and mCherry signals for indicated strains. Each dot represents one biological repilcate. Center line, mean; error bars, standard error of the mean. Statistical tests were performed by GraphPad Prism via a 1-way Anova combined with a Tukeys post-test to assess significance of the differences. * refers to a *p* value of 0.0403; ns, not significant *p* value: 0.2013. Source data are provided as a Source data file. **d** Cells incubated in medium lacking nitrogen to induce glycolipid biosynthesis were imaged 4 h after inoculation. GFP-Mac1 was expressed under control of its endogenous promoter and the fluorescence signal is depicted in cyan. For insets single channels and merged channels are shown. mCherry-SKL, magenta. Scale bars: 5 μm. **e** Cells were analyzed by super-resolution microscopy using SIM. Representative images of cells expressing GFP-Mac1 (cyan) and mCherry-SKL (magenta) (left panel), GFP-Mac3 and mCherry-SKL (middle panel), and GFP-SKL and mCherry-SKL (right panel). Full images are shown as overlays of two channels. For insets single channels and merged channels are depicted. Scale bars: 5 μm. **f** 3D-reconstruction (x,y) of GFP-Mac3 and GFP-SKL (cyan) containing peroxisomes, mCherry-SKL (magenta). Scale bars: 0.5 μm.

observed (Fig. 2g). Accordingly, the TIIV motif is necessary and sufficient for subdomain formation.

In silico analysis using the Aggrescan tool[48] predicts that TIIV may support the self-assembly of multimeric condensates (Fig. 2h). This was supported by in vitro aggregation assays, where purified 6xHis-tagged Mac3 aggregated more rapidly and to a greater extent than the Mac3-$V_{570}R$ variant (Fig. 2i). 6xHis-GFP-TIIV also was more aggregation prone than GFP-Mac-TIIR (Fig. 2j). While these in vitro results apparently align well with our in vivo observations, we assume that the TIIV motif may be part of a more complex mechanism, warranting further studies. Nonetheless, our study identifies TIIV as a key motif driving the self-assembly of a protein and compartmentalization in peroxisomal foci and subpopulations.

## Mac1 and Mac3 self-assemble in detergent resistant cores that also contain urate oxidase

Early ultrastructural studies of mammalian peroxisomes uncovered core structures, which contain the enzyme urate oxidase involved in purine catabolism as one major constituent[33,34]. We determined the localization of the putative *U. maydis* urate oxidase ortholog (UMAG_00672) fused to GFP (GFP-Uox1). Like Mac proteins GFP-Uox1 accumulated in foci and segregated to a subpopulation of peroxisomes (Fig. 3a) suggesting that in *U. maydis* all three proteins are enriched in peroxisomal core structures. Co-expression of GFP-Uox1 and mCherry-Mac1 or GFP-Uox1 and mCherry-Mac3 resulted in a high degree of colocalization (Fig. 3b, Fig. S7a, b).

It was reported previously that peroxisomal core structures can sustain treatment with nonionic detergent[49]. We prepared crude organelle extracts from strains co-expressing mCherry-SKL and GFP-Mac3, mCherry-Mac3 and GFP-Uox1 or mCherry-Mac1 and GFP-Mac3. Triton X-100-treated and non-treated samples were inspected by epifluorescence microscopy. While detection of mCherry-SKL was highly sensitive to detergent, fluorescently labeled Mac1, Mac3, and Uox1 remained visible inside of punctate structures (Fig. 3c and Fig. S7c). Transmission electron microscopy in combination with immunogold labeling demonstrated focal accumulation of GFP-Mac3 in the peroxisomal lumen (Fig. 3d). Even GFP-TIIV-SKL containing foci are detergent-resistant structures (Fig. 3e). Thus, the small motif TIIV acts as a facilitator for self-assembly of detergent resistant cores.

We could identify a motif relevant for focal localization in Mac3 but not in Mac1. As both proteins occur in the same structures after detergent treatment and Mac1 does not contain a TIIV motif (Fig. S7c), we asked whether Mac1 targets cores via Mac3. We tested if GFP-Mac1 enriches in cores in Δ*mac3* cells and mCherry-Mac3 enriches in cores in Δ*mac1* cells. In both experiments, we detected detergent-resistant fusion proteins suggesting that both proteins localize to cores independently of each other (Fig. S7d).

## Core formation permits efficient metabolism of fatty acids in the presence of Mac3

We reasoned that core structures could facilitate metabolic compartmentalization of peroxisomes. First, we examined the activity

of GFP-Mac3 and GFP-Mac3-$V_{570}R$ in vivo. To this end, we isolated MELs synthesized by Mac3 from nitrogen starved cells[40,50], expressing either GFP-Mac3 or the GFP-Mac3-$V_{570}R$ variant. Composition and amount of MELs was determined by thin layer chromatography (TLC) and liquid chromatography–mass spectrometry (LC–MS) (Fig. 4a, b). Expression of both enzymes resulted in production of similar MEL quantities with nearly identical variant distribution, demonstrating that it is not glycolipid biosynthesis that benefits from core formation.

Acyl-CoA oxidase Aox1–a typical β-oxidation enzyme–did not enrich in cores over time (Fig. 2a). The same was true for two other acyl-CoA oxidases probably involved in β-oxidation[46] that we tested (Fig. 4c). We hypothesized that selective accumulation in cores might separate enzymes not involved in β-oxidation such as Mac proteins or urate oxidase from β-oxidation enzymes to facilitate efficient fatty acid breakdown. We, therefore, compared growth of cells expressing GFP-Mac3 or GFP-Mac3-$V_{570}R$ on medium with oleic acid as sole carbon source. Cells expressing GFP-Mac3 showed a different colony appearance on solid media compared to cells expressing GFP-Mac3-$V_{570}R$, despite both fusion proteins being expressed at similar levels (Fig. 4d–f). Magnified views of colony edges revealed pronounced filamentation in GFP-Mac3-$V_{570}R$ cells, a typical stress phenomenon observed in *U. maydis*[51].

To assess if peroxisomal metabolism is indeed affected by expression of GFP-Mac3-$V_{570}R$ we measured oxygen consumption of liquid cultures challenged with oleic acid as sole carbon source. GFP-Mac3-$V_{570}R$ cells showed lower initial $O_2$ consumption rates when shifted from glucose to oleic acid in comparison with the control. The acceleration of $O_2$ consumption also differed significantly, indicating slower growth and/or delayed metabolic adaptation (Fig. 4g). $O_2$ consumption in medium containing glucose was similar for both strains (Fig. S8). Accumulation of Mac3 in cores, hence, enables luminal compartmentalization to support efficient peroxisomal metabolism.

## Proteins with diverse functions enrich in peroxisomal cores

Could accumulation of other enzymes also permit metabolic compartmentalization? To explore this, we investigated urate oxidase Uox1 core formation in a similar approach as for Mac3. Although Uox1 lacks TIIV, it contains a similar motif, TRIV, which was sufficient to drive enrichment of GFP in cores (Fig. S9a). Exchange of the valine residue at position 124 in the full-length protein to an arginine residue completely altered the localization of GFP-Uox1 and disrupted its ability to form cores in vivo (Fig. 5a). This was further validated by purifying 6xHis-tagged-Uox1 and 6xHis-tagged-Uox1-$V_{124}R$ followed by aggregation assays, as described for the Mac3 variants. The results were consistent: Uox1-$V_{124}R$ exhibited slower and less aggregation (Fig. 5b). Elevated aggregation ability was also observed for purified GFP-TRIV if compared to GFP-TRIR (Fig. S9b).

These findings again demonstrate that in vivo assembly coincides with the in vitro aggregation ability of the proteins. For Uox1 and Mac3 this requires the TIIV-like motifs.

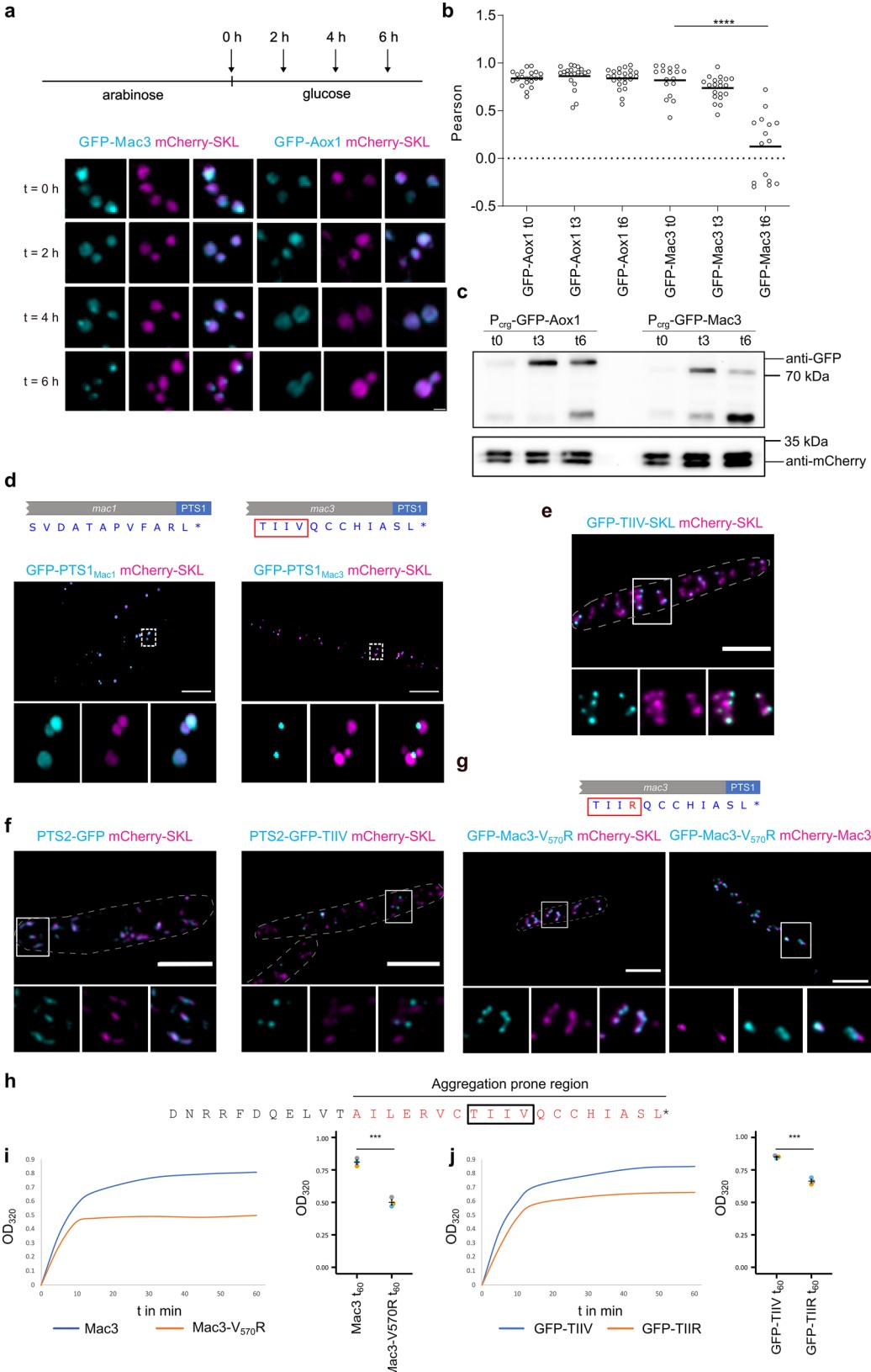

Furthermore, cells containing the altered Uox1-$V_{124}$R variant phenocopied strains with GFP-Mac3-$V_{570}$R (Fig. 5c, d). Both fusion proteins were expressed in similar amounts (Fig. 5e). For the more uniformly distributed variant Mac3-$V_{570}$R we could confirm that it was functional in vivo (Fig. 4a, b). To check if Uox1-$V_{124}$R also retained its enzymatic activity we made use of the purified 6xHis-

tagged version and measured oxidation of urate in vitro (Fig. 5f). Both tested enzyme versions showed similar conversion rates indicating that it is not the loss of enzymatic function of Uox1-$V_{124}$R that is responsible for the phenotype we observe upon incubation with oleic acid. In addition, compared to GFP-Uox1, expression of GFP-Uox1-$V_{124}$R resulted in reduced $O_2$ consumption of cells upon

**Fig. 2 | A short peptide motif enhances self-assembly to drive focal enrichment after import. a** Scheme of the transcriptional pulse-chase experiment. Cells were analyzed at indicated time points by super-resolution microscopy using SIM. GFP-Mac3 was compared to GFP-Aox1 (cyan) in strains containing mCherry-SKL. Scale bar: 0.5 μm. **b** Quantification of colocalization data derived from epifluorescence microscopy depicted in Fig. S3b based on three independent biological replicates. Samples were analyzed directly after (t0), 3 h after (t3), and 6 h after glucose addition (t6). Quantifications show Pearson's correlation coefficients of GFP and mCherry signals for indicated strains. Center line, median. Significance assessed by an unpaired, two-sided Student's *t*-test. **** refer to a *p* value lower than or equal to 0.0001. **c** Western blot analysis following the stability of GFP-Mac3 and GFP-Aox1 in a time course experiment. **d** C-terminal 12 amino acids containing the PTS1 motifs. Cells expressing GFP tagged variants were analyzed by SIM. Representative images of cells are shown. Full images are shown as overlays of two channels. For insets single channels and merged channels are shown. Scale bars: 5 μm. **e** GFP fused to

Thr-Ile-Ile-Val (TIIV) and the PTS1 Ser-Lys-Leu (SKL) was co-expressed with mCherry-SKL. Scale bars: 5 μm. **f** Cells expressing PTS2-GFP reporter proteins either with or without a C-terminal TIIV motif (cyan) were co-expressed with mCherry-SKL. Quantification showing Pearson's correlation coefficients of GFP and mCherry signals in Fig. S5e. Scale bars: 5 μm. **g** Mac3-$V_{570}$R was tagged with GFP (cyan) and localization was analyzed by epifluorescence microscopy. Quantifications in Fig. S6d. Scale bars: 5 μm. **h** The TIIV motif of Mac3 is embedded in an aggregation prone region. Aggregation of 6xHis-tagged (**i**) versions of Mac3 and (**j**) versions of GFP-TIIV and GFP-TIIR was measured over time. Plotted are the means of three replicates for each protein. Different colors denote single experiments. Center line, mean; error bars; standard error of the mean. Significance assessed with an unpaired, two-sided Student's *t*-test. *** refer to a *p* value of 0.0003 (**i**) and (**j**). Source data underlying **b**, **c**, **i**, **j** are provided as a Source data file. GFP, cyan; mCherry, magenta.

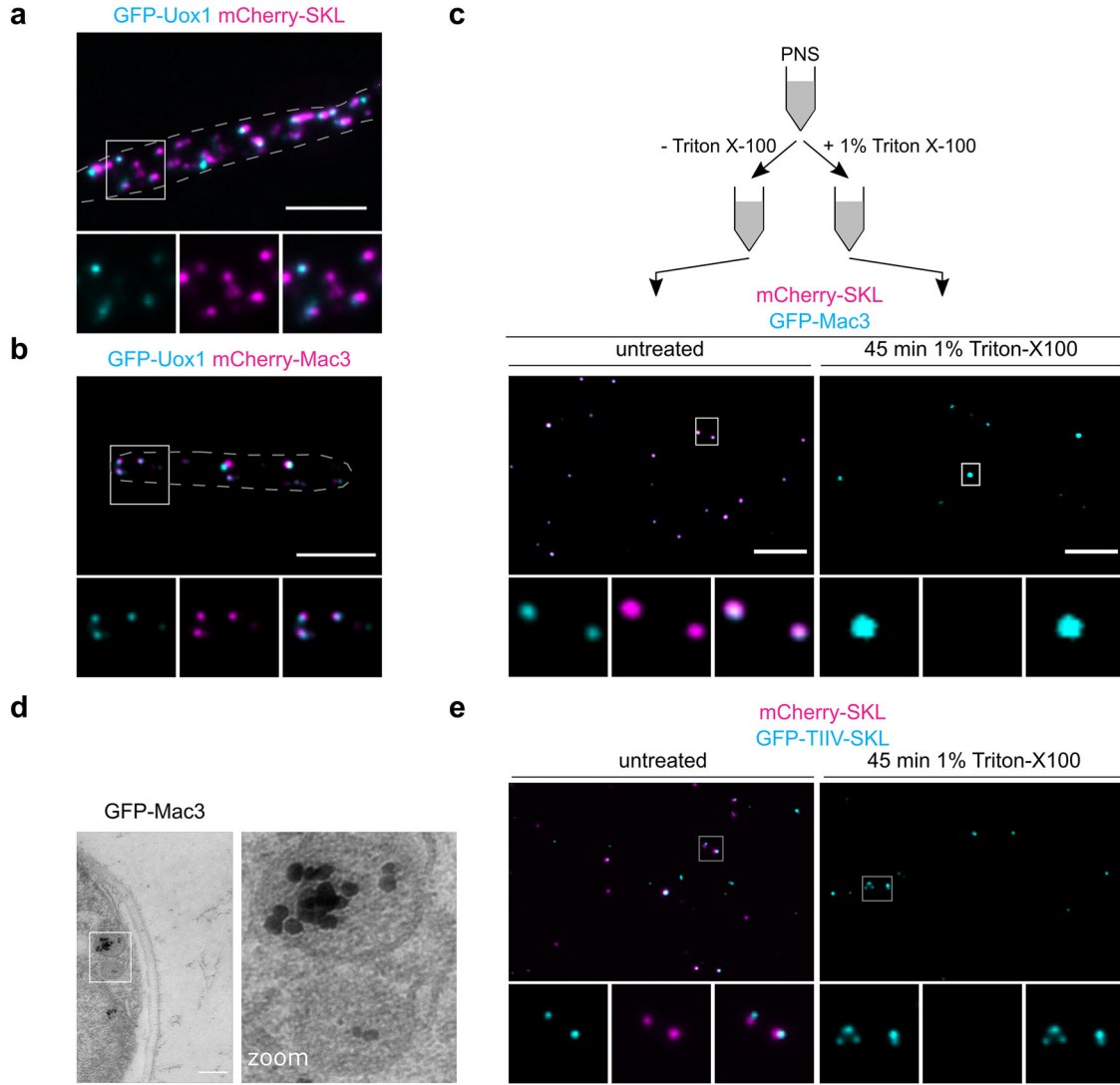

**Fig. 3 | Mac1 and Mac3 self-assemble in detergent resistant cores also containing urate oxidase Uox1. a** Representative epifluorescence images of cells expressing GFP-Uox1 (cyan) and mCherry-SKL (magenta). Full images are shown as overlays of two channels. For insets single channels and merged channels are depicted. Scale bars: 5 μm. **b** GFP-Uox1 and mCherry-Mac3; organization of figure as described above. Scale bars: 5 μm. **c** Crude organelle preparations of indicated strains were analyzed by epifluorescence microscopy after incubation in lysis

buffer (left) or lysis buffer supplemented with Triton X-100 (right) for 45 min. Organization of pictures as described above. Scale bars: 5 μm. **d** Transmission electron micrograph of cells expressing GFP-Mac3. Staining was achieved through labeling of GFP-Mac3 with anti-GFP and subsequent immunogold labeling followed by silver enhancement. Scale bar: 0.2 μm. **e** The experiment was performed as in (**c**) for the indicated reporter proteins. Scale bars: 5 μm.

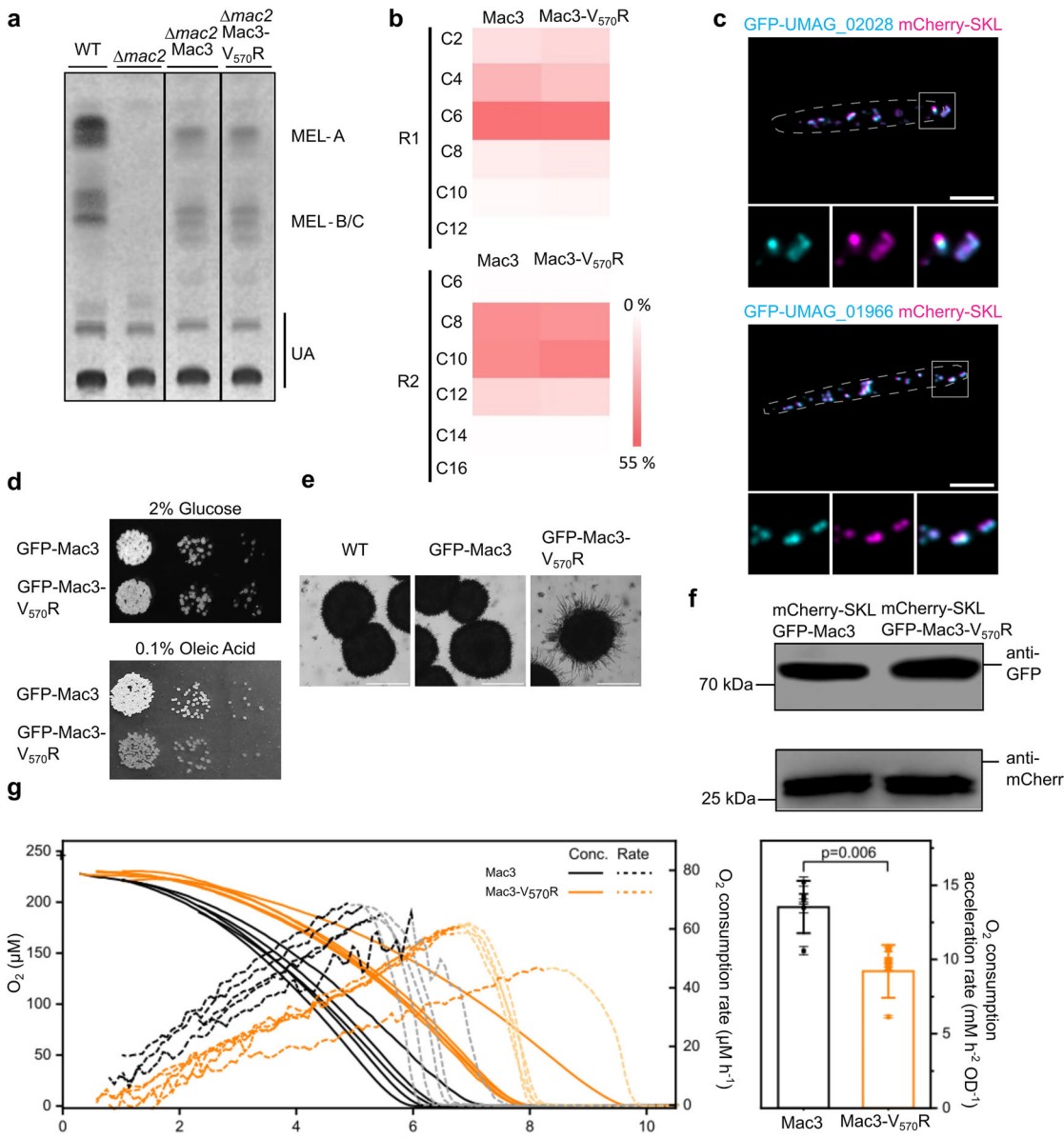

**Fig. 4 | Core assembly enables metabolic compartmentalization of peroxisomes. a** MELs produced from indicated strains were analyzed by TLC. Different MEL variants, which differ in their acetyl-moieties are specified MEL-A–MEL-C. Ustilagic acids (UAs) are a different class of glycolipids produced by *U. maydis* but not synthesized in peroxisomes[39]. Mac2 is another acyltransferase involved in MEL production, which has overlapping functions with Mac3 and was deleted to facilitate analysis. **b** Heat maps derived from LC-MS analysis of MELs produced by indicated strains. **c** Representative epifluorescence images of cells expressing GFP-fused to two additional paralogs of acyl-CoA oxidases (cyan) and mCherry-SKL (magenta). Full images are shown as overlays of two channels. For insets single channels and merged channels are depicted. Scale bar: 5 μm. **d** Serial dilutions of indicated strains were spotted on minimal media containing either glucose or oleic acid as carbon source. **e** Magnifications of colonies grown on oleic acid medium show aberrant colony morphology upon expression of the mutated version. Scale bars: 5 μm. **f** Immunoblot showing expression levels of GFP-Mac3 and GFP-Mac3-$V_{570}R$. Uncropped blots are provided in the Source data file. **g** Oxygen consumption of cells shifted from glucose to oleic acid. Oxygen concentration was recorded over time and rates were calculated as the first derivative (left). Based on the linear regression of the increase in rates, the oxygen consumption acceleration was calculated (right). Error bars on the data points denote the regression error, while the error bars on the column denote standard deviation of the mean. Significance was assessed with an unpaired, two-sided Student's *t*-test. Each line in the graph represents one biological replicate. Source data are provided in the Source data file.

incubation in oleic acid medium (Fig. 5g), again mirroring the behavior of Mac3 (Fig. 4g). Thus, two completely different enzymes that share the ability to enrich in peroxisomal cores disturb peroxisomal function if uniformly localized in the organelle lumen.

We hypothesized that core assembly could be a common principle to enable metabolic compartmentalization of the peroxisomal lumen. We therefore asked how diverse the core structures are and tested localization of additional proteins containing a predicted PTS1 (Supplementary Data S1). Six out of seven proteins with

a TIIV-like motif were enriched in core structures, but several proteins without such a motif showed focal localization as well. This suggests more redundant motifs for assembly (Fig. 5h and Fig. S10a–c). The ability to segregate into cores seems to be a feature of many peroxisomal proteins rather than an exception. These proteins include a predicted D-amino acid oxidase Dao1 (UMAG_05703) but also malate synthase Mls1 (UMAG_15004), an enzyme involved in glyoxylate metabolism (Fig. 5i). Interestingly, the signal of Mls1 did not entirely overlap with the signal of

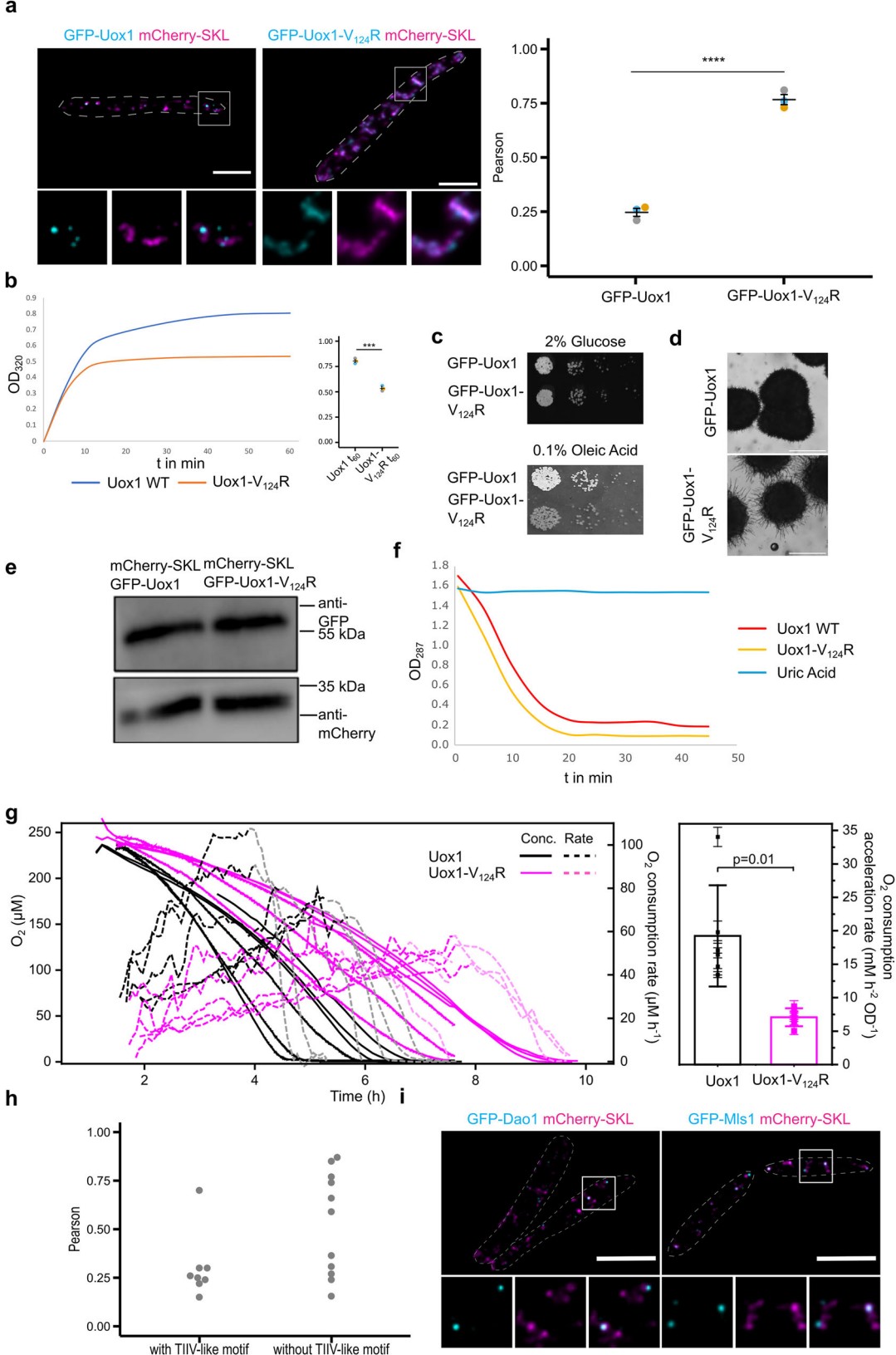

Mac3 suggesting the presence of distinct subdomains (Fig. S10d), but clarification of this phenotype requires further investigation. D-amino acid oxidase and urate oxidase are both constituents of cores found in mammalian peroxisomes[35] pointing to an evolutionary conserved phenomenon.

## Core formation properties are evolutionary conserved

Next, we asked if principles underlying core formation are phylogenetically conserved and investigated the localization of murine urate oxidase *Mm*Uox1 fused to GFP in *U. maydis* cells either expressing mCherry-SKL or mCherry-Mac3. Again, we detected a higher degree of

**Fig. 5 | Diversity and function of peroxisomal cores. a** Representative epifluorescence images of cells expressing GFP-Uox1 or GFP-Uox1-V$_{124}$R and mCherry-SKL. Full images are shown as overlays of two channels. For insets single channels and merged channels are depicted (left). Scale bars: 5 μm. Quantifications show Pearson's correlation coefficients of GFP and mCherry signals (right). Each dot represents one biological replicate. Center line, mean; error bars, standard error of the mean. Significance was assessed with an unpaired, two-sided Student's *t*-test. **** refer to a *p* value lower than 0.0001. **b** Aggregation of 6xHis-tagged versions of Uox1 was measured over time. Plotted are the means of three replicates. Quantifications show OD$_{320}$ after 60 min of aggregation. Each dot represents one biological replicate. Center line, mean; error bars, standard error of the mean. Significance was assessed with an unpaired, two-sided Student's *t*-test. *** refer to a *p* value of 0.0002 (right). **c** Serial dilutions were spotted on minimal media containing glucose or oleic acid. **d** Magnifications of colonies grown on oleic acid medium show aberrant colony morphology upon expression of the mutated

version. Scale bars: 500 μm. **e** Immunoblot showing expression levels of Uox1 variants. **f** Activity assay for Uox1 variants. **g** Oxygen consumption of cells shifted from glucose to oleic acid. Based on the linear increase in consumption rates (left), the oxygen consumption acceleration rate was plotted (right). Error bars on data points denote the regression error, while the error bars on the column denote standard deviation of the mean. Significance was assessed with an unpaired, two-sided Student's *t*-test. **h** Quantification shows the mean value of Pearson's correlation coefficients of GFP signals and mCherry signals. Quantifications for three biological replicates for each of the analyzed candidate proteins are shown in Fig. S10a–c. **i** Two candidate proteins—D-amino oxidase 1 (Dao1) and malate synthase 1 (Mls1)—were co-expressed with mCherry-SKL. Representative epifluorescence images are shown. Full images are provided as overlays. For insets single channels and merged channels are depicted. Scale bars: 5 μm. Source data underlying **a**, **b**, **e**, **g**, **h**, **i** are provided in the Source data file. GFP, cyan; mCherry, magenta.

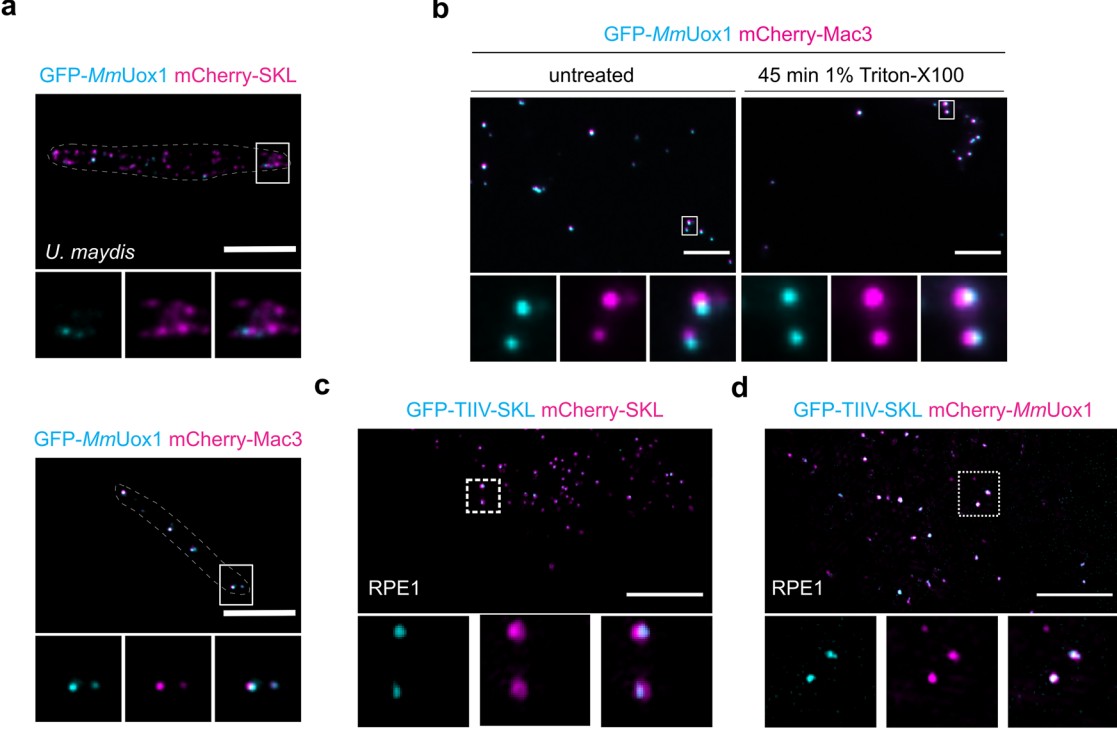

**Fig. 6 | Conservation of core formation properties. a** *Mus musculus* urate oxidase (*Mm*UOX1) was tagged with GFP (cyan) and expressed in *U. maydis* cells either co-expressing mCherry-SKL (magenta) or mCherry-Mac3 (magenta). **b** Crude organelle preparations of indicated strains were imaged by epifluorescence microscopy after incubation in lysis buffer supplemented with Triton X-100 (right). Preparations incubated in lysis buffer without Triton X-100 served as control (left). **c** Constructs expressing GFP-TIIV-SKL and mCherry-SKL were transfected into

RPE1 cells. Cells were analyzed by SIM. The representative picture shows an overlay of the GFP-signal (cyan) and the mCherry-signal (magenta) in the overview. The signal of single channels as well as the overlay are depicted for magnified insets. **d** *M. musculus* urate oxidase (*Mm*Uox1) was tagged with mCherry (magenta) and co-expressed with GFP-TIIV-SKL (cyan) in RPE1 cells. Cells were analyzed by SIM. Representative pictures are organized as above. Scale bars: 5 μm.

overlap for GFP-*Mm*Uox1 and mCherry-Mac3 compared to GFP-*Mm*Uox1 and mCherry-SKL (Fig. 6a). In addition, we could observe stability of GFP-*Mm*Uox1 containing structures upon detergent treatment of crude organelle preparations (Fig. 6b).

These data suggest that formation of core structures in peroxisomes follows analogous principles in distantly related species. To validate, GFP was fused to TIIV-SKL and this construct was expressed in human retinal pigment epithelial 1 (RPE1) cells together with mCherry-SKL. Subcellular localization of fluorescent proteins was analyzed by SIM. As in *U. maydis*, we observed accumulation of GFP-TIIV-SKL in defined subdomains of peroxisomes in RPE1 cells (Fig. 6c). To test if these structures are also enriched for a protein previously detected in the native paracrystalline core of mammalian peroxisomes we co-expressed GFP-TIIV-SKL together with mCherry-tagged *Mm*Uox1. GFP-

TIIV-SKL showed a profound colocalization with mCherry-*Mm*Uox1 revealing that both proteins are located in the same region of the peroxisome (Fig. 6d). Even the HexA protein from *Aspergillus nidulans*, which is the major crystalline component of peroxisome derived Woronin bodies[21] in filamentous ascomycetes, colocalizes with Mac3 upon heterologous expression in *U. maydis* and is enriched in detergent resistant structures (Fig. S11). Woronin bodies may hence be regarded as a more specialized evolutionary variant of cores.

## Discussion

Peroxisomal cores are stable subdomains known since the discovery of these organelles but their biological function has remained elusive. We now provide evidence that such cores enable luminal compartmentalization of individual peroxisomes and eventually the formation of

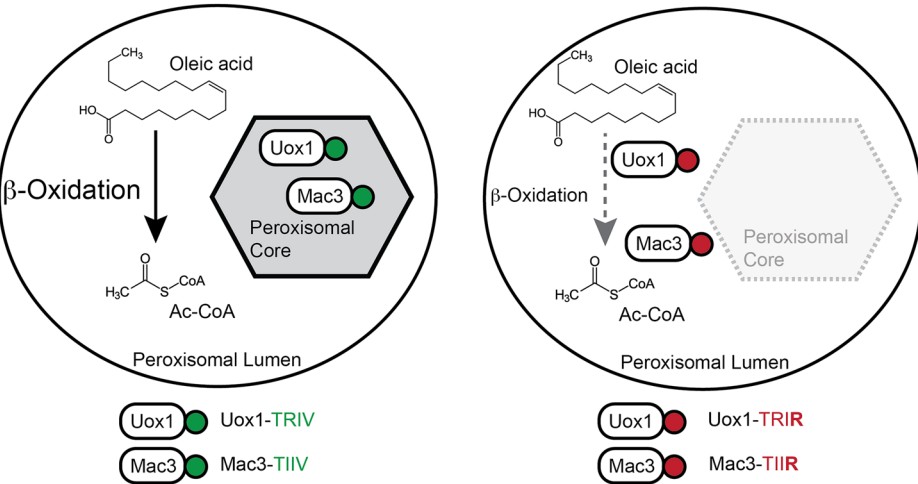

**Fig. 7 | Working model—peroxisome cores may serve for metabolic compartmentalization.** Uox1 and Mac3 assemble in detergent resistant cores, while several β-oxidation enzymes are distributed homogeneously in the entire peroxisomal lumen (left). Variation of motifs for self-assembly affected localization of Uox1 and Mac3 and growth of cells challenged with oleic acid medium (right).

peroxisome subpopulations that differ in their protein composition. Upon perturbation of core forming properties in very distinct proteins (Uox1 and Mac3), peroxisome function—presumably fatty acid oxidation—was perturbed (Fig. 7).

Within the cores of *U. maydis* we detected an accumulation of enzymes with diverse functions, excluding key enzymes for fatty acid oxidation (Fig. 5 and Fig. S10). Core forming proteins include enzymes involved in secondary metabolism (Mac1 and Mac3) and enzymes that produce $H_2O_2$ (Dao1 and Uox1). We also detect less well characterized enzymes e.g., an epoxide hydrolase, which contains a TIIV-like motif in *U. maydis* and in mice (Supplementary Data S2) and may play a role in detoxification of xenobiotics[52,53].

Interestingly, orthologs of Dao1 and Uox1 have already been detected in peroxisomal cores of mammalian cells[34,54] suggesting a functional relevance for segregation of specific enzymatic functions in distantly related species. For the $H_2O_2$ producing enzymes a peroxisomal localization is intuitive, as peroxisomes detoxify this molecule by catalase. Also $H_2O_2$ producing methanol oxidase of several yeast species is enriched in crystalloids[55–57]. Accordingly, enzymes localizing in peroxisomes due to their oxidative function, but unrelated to fatty acid oxidation are more regularly enriched inside cores.

Previously, others have identified subpopulations of rat liver peroxisomes that differ in protein content[25,58]. The different populations of peroxisomes contain an altered ratio of peroxisomal acyl-CoA oxidase to urate oxidase—a typical core protein in this tissue[25,58]. These findings are in agreement with ours and hint on an evolutionary conserved phenomenon—formation of peroxisome subpopulations may more generally coincide with core formation.

The biophysical properties of *U. maydis* core structures are still unknown. We observe condensed structures by TEM (Fig. 3d). If these are crystalline structures as observed for urate oxidase[32] and methanol oxidase[55] or if they show distinct characteristics as described for d-amino acid oxidase condensates of kidney cells[59] remains to be elucidated. Here, we demonstrate that proteins identified as core residents in other systems, such as murine urate oxidase and the Woronin body protein HexA, exhibit focal accumulation within *U. maydis* peroxisomes (Fig. 6 and Fig. S11). Their localization closely resembles native core proteins, suggesting similar mechanistic principle underlying assembly. The identification of TIIV as functional but redundant motif required for this assembly is a valuable starting point to gain more insights into the underlying molecular-mechanistic principles. Our biochemical experiments demonstrate that variation of the motifs in Uox1 and Mac3 affects their aggregation ability in a very

similar way. Such elements could be involved in initial association during the assembly process (Figs. 2 and 5) e.g., via dimerization. Other features of core forming proteins may then determine biophysical characteristics of the subdomains as well as their individual composition. Knowledge about the motifs helped us to investigate proteins that mis-localize into the peroxisomal lumen via a minimal change of the primary sequence still retaining their enzymatic activity. Their expression caused a growth phenotype upon challenge with oleic acid (Figs. 4 and 5). It will be interesting to learn if such an effect also occurs upon turning other candidate core proteins (Fig. 5) more soluble and thereby inhibiting luminal compartmentalization.

Our understanding of functionally different luminal peroxisomal compartments as well as of peroxisome subpopulations is only at the beginning. Peroxisomes are simple vesicular structures and typically show no internal membranes. A notable exception are more recently described intraluminal vesicles of plant peroxisomes, which represent a different approach for metabolic compartmentalization[60]. For mitochondria internal compartmentalization is much more pronounced and in part achieved via a complex inner membrane system[61,62]. Furthermore, functionally different subpopulations for this organelle were described by others[63]. Metabolically distinct mitochondrial species were very recently discovered in mammalian cells. They either enrich machinery for ATP production or proline biosynthesis enzymes—segregation into subpopulations involves multimerization of pyrroline-5-carboxylate synthase[64]. Thus, the emergence of metabolically different organelle subpopulations via emergence of stable condensates could be a general feature of organelles.

For many years a thorough characterization of the peroxisomal proteome was a major focus of research revealing a plethora of biochemical pathways[65]. Selective core formation appears to be one clue how the simple single-membrane bounded peroxisomes can perform a fascinating potpourri of metabolic tasks[2,6,66–69].

## Methods

### Microorganisms, growth conditions, and transformation

*Escherichia coli* strain TOP10 (Invitrogen) was used for all cloning purposes and amplification of plasmid DNA. *U. maydis* strains generated and used in this study are derivatives of Bub8, MB215 and Bub8 mCherry-SKL[19,50,70] and listed in Supplementary Data 3. All strains are available from the authors upon reasonable request. *U. maydis* cells were grown at 28 °C in YEPSlight medium[71] or yeast nitrogen base medium (YNB; Difco) at pH 5.6 supplemented with 0.2% (w/v) ammonium sulfate either containing 2% (w/v) glucose (YNB-G) or a

mixture of 0.2% (v/v) oleic acid (Roth)/0.1% (v/v) Tween-40 (YNB-O) as a carbon source. For epifluorescence imaging cells were grown in oleic acid media if not specified differently. SIM imaging was performed on glucose grown cells to circumvent artifacts of oleic acid addition. For chase experiments *U. maydis* cells were grown overnight in YEPSlight medium. Subsequently, cells were washed and diluted in YNB medium containing 2% (w/v) arabinose and 0.2% (w/v) ammonium sulfate and incubated for 2 h. Expression of fusion proteins was stopped by shifting the cells to YNB medium containing 2% (w/v) glucose and 0.2% (w/v) ammonium sulfate and the localization of the fluorescent proteins was microscopically monitored at indicated time points. Cell transformation was performed as previously described[72,73]. DNA was integrated into the *ip*-locus of *U. maydis* cells[57] or randomly integrated into the genome. Hygromycin (200 μg/ml), carboxin (2 μg/ml), and G418 (400 μg/ml) were used for selection. Genomic DNA was extracted as previously described[74]. To assess growth, serial dilutions of cells were spotted on solid YNB-G or YNB-O media and growth was assessed after 2–5 days. Colonies were magnified with a binocular microscope.

### Human cell line and transient transfection of cells
hTert-RPE1 cells (ATCC CRL-4000) were cultured in DMEM-F12 medium (Life Technologies) supplemented with 10% (v/v) FCS and Penicillin/Streptomycin (100 U/mL penicillin, 100 μg/mL streptomycin) at 37 °C, 5% $CO_2$. Cells were transiently transfected with the help of Fugene 6 (Promega) using a ratio of 3:1 of reagent to DNA. Cells were transfected in MatTek 3.5 cm glass-bottom dishes. 12 h after transfection, the medium was replaced to remove transfection reagent, and the cells were imaged between 16 h and 20 h post transfection.

### Molecular cloning and nucleic acid procedures
Standard procedures were followed for plasmid generation[75]. Plasmids used for protein expression in *U. maydis* are derivatives of the plasmids pOTEF-GFP-Ala6-MMXN and pCRG-GFP-Ala6-MXN[76]. Plasmids used for expression of fluorescent proteins in hTert-RPE1 cells are derivatives of the plasmid pEGFP-C1 and pmCherry-C1 (Clontech). Plasmids were analyzed by sequencing. All plasmids and oligonucleotides used in this study are listed in Supplementary Data S3.

### Epifluorescence microscopy
Logarithmically grown *U. maydis* cells were treated with 100 μM Carbonyl cyanide m-chlorophenyl hydrazine (CCCP) resolved in dimethyl sulfoxide to block peroxisomal movement. For the majority of experiments using epifluorescence microscopy cells were incubated in oleic acid-containing media for 4 h prior to imaging experiments as formation of subpopulations was easier to follow under these conditions (exceptions: imaging experiments underlying Fig. 1a and Fig. S5). After 5–10 min incubation at room temperature, the cells were either placed on 1% (w/v) agarose cushions containing 100 μM Carbonyl cyanide m-chlorophenyl hydrazine CCCP on a microscope slide and covered with a coverslip or placed on MatTek 3.5 cm glas-bottom dishes and covered with an 1% (w/v) agarose cushion containing 100 μM CCCP as well. For time-lapse imaging no CCCP was added. Epifluorescence microscopy was performed on an Axiovert 200 M inverse microscope (Zeiss) equipped with a 1394 ORCA ERA CCD camera (Hamamatsu Photonics), filter sets for eGFP and rhodamine, and a Zeiss 63x Plan Apochromat oil lens (NA 1.4). The software Volocity 5.3 (Perkin-Elmer) was used for image acquisition. The ImageJ plugin DeconvolutionLab with 25 iterations of the Richardson-Lucy algorithm was used for image deconvolution, point-spread functions for each channel were computed by the ImageJ plugin PSF Generator using the Richards & Wolf model. Alternatively, epifluorescence microscopy as well as time-lapse experiments (500 ms time-lapse for 60 s) were performed on a Deltavision OMX V4 microscope (GE Healthcare) equipped with three water-cooled PCO edges CMOS cameras, a solid-state light source and a laser-based autofocus. The

software softWoRx (Applied Precision, GE Healthcare) was used for image deconvolution. Images were processed in ImageJ/Fiji[77].

### Structured illumination microscopy
Three-dimensional structured illumination microscopy was performed on a Deltavision OMX V4 microscope with an Olympus x60 NA 1.42 objective and three PCO edges CMOS cameras and 488 nm and 568 nm laser lines. Cells were illuminated with a grid pattern. 15 raw images (3 orientations, 5 phases) were taken for each image plane. SoftWoRx software (Applied Precision, GE Healthcare) was used for image reconstruction, alignment, and projection. Images were processed in ImageJ/Fiji[77]. For 3D visualization of peroxisomal subdomains, surfaces were generated using Imaris (Oxford instruments). Imaris segments objects and then generates 3D surfaces/3D meshes from these objects which are then visualized as 3D objects. The "hollow" ring like appearance is based on the visualization of the surface in the pictures shown in Fig. 1f and S2g, a partially transparent texture is applied. Alternatively, SIM images were acquired by a Zeiss ELYRA PS.1 microscope with an ANDOR iXon 987 EMCCD camera (exposure time: 50 ms, gain: 5) using an alpha Plan-Apochromat 100x/1.46 Oil DIC M27 Elyra objective with 488 nm (HR Diode 488-200, 3%) and 564 nm (HR DPSS 561-200, 5%) laser lines as excitation sources. 15 images (3 rotations, 5 phases) were collected per plane. The software ZEN (Zeiss) was used for processing the SIM reconstructions.

### Transmission electron microscopy and immunogold labeling
For visualization of *U. maydis* cells expressing GFP-Mac3 using transmission electron microscopy (TEM) 50 ml logarithmically growing cells were harvested and frozen under high-pressure (Wohlwend HPF Compact 02). After subsequent freeze substitution (using acetone, containing 0.25% (w/v) osmium tetroxide, 0.2% (w/v) uranyl acetate, 0.05% (w/v) ruthenium red, and 5% (v/v) water) (Leica AFS2), cells were embedded in Epon812 substitute resin (Fluka). Embedded cells were sectioned to 50 nm thin sections (Leica EM UC7 RT), which were used for immunolabeling with anti-GFP (Rockland; dilution 1:500). As a secondary antibody rabbit-anti goat antibody coupled to ultra-small gold particles was used (Aurion, dilution 1:100). Subsequently a silver enhancement procedure was performed and sections were post-stained with 2% (w/v) uranyl acetate and 0.5% (w/v) lead citrate. Analysis of the samples was conducted with a JEOL JEM2100 TEM equipped with a fast-scan 2k CCD TVIPS (Gauting, Germany) F214 camera. Images were processed in ImageJ[77].

### Preparation of crude organelles
Preparation of post-nuclear supernatants was adapted from previous protocols[78–80]. Briefly, 200 ml of logarithmically growing cells were harvested for 5 min at 1600 × g at 23 °C. Cells were washed two times with deionized water and two times with sorbitol buffer (1.2 M sorbitol; 20 mM $KH_2PO_4$ adjusted to pH 7.4 with $K_2HPO_4$). Formation of spheroplasts was achieved through incubation in sorbitol buffer containing novozyme (4 μg/ml) for 30–45 min. Degradation of the cell wall was followed microscopically. Afterwards, spheroplasts were kept on ice and gently washed two times with sorbitol buffer−centrifugation for 10 min at 1000 × g rpm at 4 °C. Spheroplasts were gently washed with lysis buffer (5 mM MES, 0.5 mM EDTA, 1 mM KCl, 0.6 M Sorbitol, 1 mM 4-aminobenzamidine dihydrochloride, 1 μg/ml aprotinin, 1 μg/ml leupeptin, 1 mM phenylmethylsulfonyl fluoride, 10 μg/ml N-tosyl-L-phenylalanine chloromethyl ketone, and 1 μg/ml pepstatin)−centrifugation for 10 min at 1000 × g rpm at 4 °C, resuspended in 15 ml lysis buffer and frozen at −80 °C overnight. Homogenization was achieved through 2 × 10 strokes with a Potter-Elvehjem homogenizer interrupted by chilling the samples on ice for 2 min. Nuclei and cell debris were removed by two subsequent centrifugations at 1600 × g for 10 min. Subsequently, the PNS was diluted to an OD600 of 1, aliquoted and frozen at −80 °C.

## Detergent treatment

PNS fractions were thawed on ice, centrifuged at $10,000 \times g$ for 10 min at 4 °C and concentrated 10× in lysis buffer. Samples were split into halves and aliquots were incubated in lysis buffer containing 1% Triton-X-100 or in lysis buffer for 45 min. Preparations were immediately analyzed by epifluorescence microscopy immobilized on 1% (w/v) agarose cushions.

## Analysis of glycolipids

Extracellular glycolipids were extracted as described in Hewald et al.[81]. Tween-20 was used in a final concentration of 0.05% (v/v). Glycolipids were separated by TLC on silica plates first with a solvent system consisting of chloroform-methanol-water (65:25:4, v/v) for 4 min followed by two times with a second solvent system consisting of chloroform-methanol (9:1, v/v) for 18 min each[82]. Plates were air-dried and sugar containing compounds were visualized by application of a mixture of ethanol: sulfuric acid: p-anisaldehyde (18:1:1, v/v) followed by heating at 150 °C for 2 min[83].

## HPLC

High-performance liquid chromatography (HPLC) separation of the extracted MELs (50 µl) was performed with a 1100-HPLC system (Agilent) equipped with an EC 125/2 Nucleodur 100-3 C8 ec column (Macherey-Nagel). The gradient was applied at a flow rate of 0.2 ml/min and a column temperature of 45 °C. Conditions were as follows (buffer A is water with 0.05% (v/v) formic acid; buffer B is methanol with 0.045% (v/v) formic acid): linear gradient from 60% buffer B to 95% buffer B within 30 min and followed by constant flow in 95% buffer B for 10 min.

## Mass spectrometry−electrospray ionization

Online electrospray ionization MS and $MS^n$ of the HPLC-separated compounds was done with a Finnigan LTQ-FT Ultra fourier transform ion cyclotron resonance (FT-ICR) mass spectrometer (Thermo Fisher). Electrospray ionization parameters were adapted to the flow rate and mass range. Accurate masses (accuracy 2 ppm or better), allowing the determination of the chemical formulas of the eluting compounds, were obtained by using the FT mass analyzer at a resolution of 100.000. Meanwhile, fragment ions were generated and analyzed in the LTQ mass analyzer. Alternatively, data-dependent fragmentation (untargeted) or fixed m/z fragmentation (targeted) was used; whereby the latter resulted in better signal to noise ratios and sensitivity. The accurate FT masses in combination with $MS^2$ experiments were sufficient to identify the acylation pattern of the compounds. Data were analyzed using the software XcaliburTM (Thermo Fisher). Heat maps were generated using Microsoft Excel.

## Heterologous protein expression in *E. coli* and purification of proteins

Open reading frames encoding Mac3, Mac3-$V_{570}$R, Uox1, Uox1-$V_{124}$R, GFP-TIIR, GFP-TIIV, and GFP-TRIV proteins were amplified by PCR and cloned into a modified pET24d vector (Novagen) using *BsaI* restriction sites. Both proteins contained an N-terminal hexa-histidine tag (6xHis). Proteins were produced in *E. coli* BL21(DE3) (Novagen) over 16 h at 30 °C in LB-medium containing 1% (w/v) lactose. Cells were lysed by a Microfluidizer (M110-L, Microfluidics). After lysis cell debris was removed by high-speed centrifugation ($50,000 \times g$). All proteins were purified by nickel-ion affinity and size exclusion chromatography (SEC), as described previously[84]. The SEC buffer consisted of 20 mM HEPES-Na (pH 7.5), 200 mM NaCl, and 20 mM KCl.

## Aggregation assay

The protein was diluted in 1 × SEC Buffer to a concentration of 2 µM (total volume 10 ml). 10 µl of 1 M DTT stock solution (final concentration 1 mM) was added. The mixture was carefully mixed on ice and transferred to room temperature for 30 min. Next, samples were incubated at 55 °C for

60 min. Samples were taken at different time points (i.e., 0, 5, 10, 15, 30, and 60 min), and analyzed at an OD of 320 nm. 1 × SEC Buffer served as blank. The data was plotted with Microsoft Excel and further analyzed as described in the statistical analysis section.

## Immunoblotting and antibodies

Small amounts of protein were extracted with a buffer consisting of 50 mM Tris-HCl, pH 6.8, 2% (w/v) SDS, 6% (v/v) glycerol, 0.025% (w/v) bromophenol blue, and 50 mM dithiothreitol. In brief, 1 $OD_{600}$ of yeast cells were centrifuged at $13,000 \times g$ for 1 min and incubated with 300 µl 0.2 M NaOH for 5 min at room temperature. Cells were centrifuged, resuspended in 50 µl sample buffer, incubated for 5 min at 95 °C, centrifuged again, and the supernatant was transferred to a new reaction tube. Proteins were incubated for 5 min at 95 °C and rotated at 750 rpm prior to loading. SDS−PAGE was performed with self-cast or Midi-protean TGX precast gels (BioRad), PageRuler Prestained protein ladder (ThermoFisher) as protein standard and a BioRad Mini- or Midi-Protean cell. Proteins were blotted on PVDF membranes in a BioRad Mini Trans-Blot cell or Criterion Blotter at 30 V for 16 h at 4 °C or at 70 V for 1 h at 4 °C. Membranes were blocked for 30 min in TBST containing 5% (v/v) nonfat dry milk and incubated in primary antibody containing 0.02% (w/v) sodium azide with gentle agitation for 2 h at room temperature or overnight at 4 °C. After removal of the antibody solution, membranes were washed five times with TBST for 5 min, incubated with HRP-conjugated secondary antibody for 45 min at room temperature, washed five times with TBST for 5 min, and then developed using either Pierce ECL Immunoblotting Substrate (ThermoFisher).The following antibodies were used: anti-GFP (1:2000; TP401, Torrey Pines Biolabs), anti-mCherry (1:1000; TA150125, ThermoFisher), goat anti-mouse IgG-HRP (1:10.000–1:50.000; 31430, ThermoFisher), goat anti-rabbit IgG-HRP (1:10.000–1:50.000; 31460, ThermoFisher).

## Oxygen consumption measurements

To estimate if the delocalization of core proteins to the peroxisomal lumen affects oleic acid metabolism, we considered that rates of oleate oxidation relate to the consumption rates of $O_2$ if oleic acid is added as sole carbon source. We therefore assessed the dynamics of oxygen consumption following the transition to oleic acid. *U. maydis* cells were grown to an $OD_{600}$ of 1 in YNB medium containing 2% (w/v) glucose and 0.5% (w/v) ammonium-sulfate. To remove glucose, cultures were washed three times in YNB. Next, an $OD_{600}$ was adjusted to 0.01 in YNB medium with 0.5% (w/v) ammonium-sulfate and oleic acid and Tween-40 were premixed and added to final concentrations of 0.2% (v/v) and 0.1% (v/v), respectively. Cultures were transferred into gas-tight septum vials (Exetainer, Labco, UK) equipped with sensor spots (Pyroscience, Germany) allowing contactless oxygen measurements in the vials through a fiberoptical read-out system (FireSting-Pro, Pyroscience, Germany). The vials were closed without headspace, and $O_2$ concentration was monitored (Pyro Workbench) while cultures were continuously stirred. Rates of oxygen consumption were calculated as the first derivative of concentration over time. Acceleration, reflecting the combined effect of growth and metabolic adaptation, was determined by linear regression of these rates over time. Analogous control experiments were performed for cells incubated in YNB containing glucose as carbon source.

$P$ values were calculated by an unpaired, two-sided Student's $t$-test.

## Urate oxidase assay

We adapted the protocol from two different sources[85,86]. Briefly, 300 µl of borate buffer (20 mM boric acid, 15 mM sodium chloride, 5 mM sodium tetraborate, pH 8.4) and 450 µl of uric acid solution (20 mg uric acid in 25 mM NaOH, pH 8.0) was added to a cuvette, mixed and measured as a starting point before protein was added. Next, 200 µl of enzyme (around 50 µM in borate buffer) was added and 30 s after addition of the enzyme the first measurement was taken. Next, the

cuvette was sealed with parafilm and placed in a water bath at 30 °C. During the reaction every 5 min a measurement was taken until no change in absorption occurred. As a control, 750 μl of buffer and 200 μl of enzyme were mixed, kept in the water bath during the reaction time and measured after 50 min, Absorbance was measured on a Ultrospec 2100 pro UV/vis spectrometer at 287 nm with a polystrol/polystyrene cuvette (10 × 4 × 45 mm). Borate buffer was used as a blank. For the uric acid measurement, 450 μl of uric acid solution were added to a cuvette containing 500 μl of borate buffer. Measurements were taken as described above.

## Quantification of data and statistical analysis

Quantifications of aggregation assays are based on three independent experiments. Microscopic data was collected from at least three independent *U. maydis* cultures (biological replicates) for each experiment. At least five images per culture with at least ten cells were quantified unless otherwise stated (Fig. 2b). Quantification of colocalization was performed as follows. Detection and measurement of individual peroxisomes was performed automatically via an ImageJ macro (Supplementary File 1). Briefly, maximum projections of deconvolved RFP and GFP z-stacks were created, transformed into 16-bit images and corrected for channel-misalignment by the "StackReg 2.0.0" plugin using "Rigid Body" transformation (https://sites.imagej.net/BIG-EPFL/plugins/)[67] to later serve as the reference images for detection of peroxisomes; same operations were performed on average projections of the respective non-deconvolved z-stacks for later fluorescence-intensity measurements. To increase sensitivity and account for peroxisomes showing only fluorescence in one of the two channels, the resulting images were merged using the Image Calculator function with the maximum option (detection based on only RFP or GFP references was still included in the macro but performed less reliable than using the merge of both references). For detection of peroxisomes, the merged average projections were used to create a mask of the cells. This mask was used to determine the background noise outside of the cells in the reference merge. The resulting maximum noise intensity was used for the Find Maxima function of ImageJ; individual peroxisomes were detected in the reference merge and marked as point selections. These selections were restored in the background-corrected average projections of the non-deconvolved stacks of both channels. Subsequently, circular ROIs of a 3-pixel radius (area of 32 pixels) were created around the point selections and signal intensities were measured for both channels (as Raw Integrated Density per ROI) and stored side-by-side in the results file for the whole dataset. A commented version of the macro is attached to the manuscript. For evaluation of the resulting data, the statistics software R was used (R Core Team, 2015): measured intensities were normalized to one and a linear regression analysis was performed with the RFP intensities as independent and GFP intensities as dependent variable, in addition a correlation test was performed (Pearson's) using Volocity 5.5.2 or R. Superplots and statistical tests were computed using RStudio 1.2.1335 with R 3.6.0 as described[87,88]. Plots are structured as follows: center line, mean; error bars, standard error of the mean; circles, mean of experiments. *P* values were calculated using an unpaired, two-sided Student's *t*-test. For data, which contain multiple comparisons a 1-way Anova combined with a Tukeys post-test was performed to assess significance of the differences. * refers to a *p* value lower than or equal to 0.05; ** refer to a *p* value lower than or equal to 0.01 and ***lower than or equal to 0.001. ****lower than 0.0001. The graph shown in Fig. 2b was generated using GraphPad Prism. Each dot represents one cell. Center line, median. Source data underlying the quantifications is provided in the Source data file.

## Bioinformatics and accession numbers

*U. maydis* putative peroxisomal PTS1 protein sequences were retrieved from the *Ustilago maydis* genome database (MUMDB) or broad institute and manipulated with notepad ++ using regular expressions (regex). PTS1 motifs were bookmarked with the regular expression "([SA]

[RKHQNS][LI]|[SA][RK][MFV]|[PCVGE][RK][LI])\*$". Protein sequences containing a TIIV-like motif were bookmarked with the regular expression "(T [I, L, V] [I, L, V] [I, L, V] [I, L, V] [I, L, V])\*$". Peroxisomal candidate proteins with TIIV-like motif and without motif were randomly chosen and the localization of the peroxisomal proteins to subdomains was analyzed by fluorescence microscopy. All genes can be accessed via the National Center of Biotechnology Information via the UMAG number.

## Statistics and reproducibility

For all data showing representative images the experiments were at least performed three times with similar results. Please note the many of the data have also been quantified and that the quantifications are shown in Supplementary Information file.

## Reporting summary

Further information on research design is available in the Nature Portfolio Reporting Summary linked to this article.

## Data availability

Source data are provided with this paper.

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

## Acknowledgements

We thank Marisa Piscator and Ulrikke Dahl Brinch for excellent technical assistance. We are grateful to Reinhard Fischer for reagents, Maya Schuldiner for advice on the project. We acknowledge Uwe Linne for providing LC-MS data. Open Access funding provided by the Open Access Publishing Fund of Philipps-Universität Marburg. J.A. was supported by a fellowship from Marburg Research Academy. T.S. acknowledges funding from SYNMIKRO. B.S. received funding from the DFG (grant ID: SA 1018/5-1). K.S. was supported by a Career grant from the South-Eastern Norway Regional Health Authority (2020038) and a Research Grant from the Research Council of Norway (315103). G.B. thanks the European Research Council (ERC) for support through the project "KIWIsome" (Grant agreement number: 101019765). J.F. acknowledges funding from the DFG (grant ID FR-3586/2-1).

## Author contributions

Conceptualization: N.B., J.A., D.M., J.M.K., K.S., G.B., and J.F.; Data acquisition: N.B., J.A., J.B., D.M., C.M., M.C., J.P., T.H., J.M.K., B.S., K.S., and J.F. Preparation of figures: N.B., J.A., J.B., D.M., K.S., J.M.K., and G.B.; Generation of code, bioinformatic and statistical analysis: N.B., J.A., C.R., M.C., T.S., J.M.K., K.S.; Contribution of material: V.W.; Supervision: C.T., M.B., K.S., G.B., J.F.; Funding acquisition: C.T., B.S., J.M.K., K.S., G.B., J.F.; Writing—original draft: J.F.; Writing—revision and editing: All authors. Analysis of data: All authors.

## Funding

## Competing interests

The authors declare no competing interests.
