## [Transparent Peer Review file · Nature Communications]

Peroxisomal core structures segregate diverse metabolic pathways

Corresponding Author: Dr Johannes Freitag

Version 0:

Reviewer comments:

Reviewer #1

(Remarks to the Author)

In this manuscript, Bäcker and colleagues looked at how and why peroxisomes may compartmentalize their protein content. The hypothesis is that the “sub-compartments” observed in peroxisomes from some organisms may provide a functional advantage by segregating metabolic pathways. This is an important subject and the work described in this manuscript has great potential to advance our knowledge on peroxisome biology. However, while the author’s hypothesis is plausible, the data supporting it needs to be improved. The authors should address the following points:

Major issues:

1- Lines 34-36, 59-64, 233-239, and 351-353 – The idea that peroxisomes from many species contain well defined functional sub-domains in their matrix should be toned down. Note that crystalline cores of UOX in rat liver contain essentially UOX and no other enzymes (e.g., PMID: 1594592). In agreement with this, ref. 35 shows that, except for UOX, all the other enzymes studied are soluble in the presence of detergent. Also, crystals of alcohol oxidase (AOX) from *Pichia pastoris/Hansenula polymorpha* can form in vivo when expressed at high levels, even in the absence of peroxisomes, suggesting that this is an intrinsic property of the protein (PMID: 27006771; PMID: 27458710 and references cited therein). Finally, the enzymatic content of rodent and human peroxisomes is almost identical with the exception that humans lack UOX (and peroxisomal crystalloids). If the function of these crystalloids were to compartmentalize non-beta-oxidation enzymes, humans would be in trouble.

2- Lines 117-118 – Conceptually, intraperoxisomal selective degradation should also be considered here. In the pulse chase experiment shown in Fig 2a there is an apparent decrease in the fluorescence of GFP-Mac3 over time. Is it possible that soluble Mac3 is constitutively degraded in peroxisomes while the aggregated Mac3 pool resists proteolysis? The authors should check Mac3 protein level over time by western blotting. Also, in Fig. S3b, at t=3h, almost all magenta and cyan dots overlap, yet at 6h the authors show a cell in which most peroxisomes have mCherry-SKL but lack GFP-Mac3. According to the explanation provided by the authors (lines 130-131), this would imply that peroxisomes more than doubled their numbers in 3 hours. But if this is true, then the fluorescence intensity of GFP-AOX1 in relation to mCherry-SKL should be decreased by a factor >2. This decrease cannot be perceived in fig. S3b. Did the authors check this? Otherwise, the different subpopulations of peroxisomes may be better explained by degradation of Mac3, and not by the unequal distribution of the aggregates during division.

3- Data in Fig. S3c - Both GFP-9mer-SKL and GFP-7mer-SKL already display puncta that do not localize precisely with mCherry-SKL. It is unclear whether these non-localizing puncta are independent organelles or subdomains of the same organelle. Please clarify. More strikingly, there is also heterogeneity in mitochondria containing MTS-GFP and MTS-mCherry (Fig. S3f) – in the “overlap” inset of that figure, one can see zones with “pure” cyan and “pure” red. The heterogeneity in this negative control is difficult to understand, when such differences are sometimes interpreted as segregation (e.g. Fig. 6 and c). Please clarify.

4- It is surprising that cells expressing either GFP-Mac3-V570R or GFP-Uox1-V124R display the same phenotype. While one can see the link between the acyltransferase and β -oxidation, it is harder to see how Uox1-V124R would affect growth in oleic acid, especially considering that other “core-forming” peroxisomal proteins should still form these structures. Is the idea that any one abundant core-forming protein will seed the aggregation/crystallization of others? The authors should

discuss their model in more detail.

Other issues:

5- Lines 47-48 – “through a hydrogel-like channel resembling the interior of the nuclear pore complex”. This is not a fact. This is simply one of many very hypothetical models presently available on how proteins translocate the peroxisomal membrane (e.g., PMID: 38936257).

6- Section “A short peptide motif in Mac3...”- please note that alphafold predicts with high confidence the structure of Mac3 (<https://alphafold.ebi.ac.uk/entry/A0A0D1EB13>). In the predicted model the TIIV motif is part of an alpha-helix and the 3 hydrophobic residues of the motif (Ile-Ile-Val) face the hydrophobic core of the protein, as expected. Thus, it is very difficult to understand how such a motif might work. Some speculations in the discussion would be welcome.

7- Lines 208-209 – the authors have not shown that *U. maydis* GFP-Uox1 forms paracrystalline structures. Thus, the conclusion in this last sentence is not supported by the data. Please rephrase.

8- Lines 96-98 – Please provide a short explanation of why the GFP-/mCherry-SKL proteins appear as rings in the 3D-reconstruction (fig. 1f) for those not familiar with these techniques

9- Fig S3f - the mitochondrial localization of MTS-GFP-TIIV should be supported with more robust data (e.g., using a mitochondrial outer membrane marker). As presented, there may be doubts on the localization of this protein (mitochondrial or cytosolic?)

10- Some figures are misidentified in the main text:

- Fig.S3 c to i are misidentified as Fig.S2 at several places
- In line 198, S3a should be S4a
- In line 204 S3b should be S4b
- In line 258, e should be f
- Figure S7 and S6 are swapped.

Reviewer #2

(Remarks to the Author)

Dear editors,

the manuscript „Peroxisomal core structures segregate diverse metabolic pathways” submitted by Freitag and colleagues reports the discovery of a phylogenetically conserved mechanism separating peroxisomal matrix proteins into homogeneously distributed enzymes or centrally concentrated core structures. The results presented are primarily based on experiments performed in the filamentous fungi *Ustilago maydis*, in which the authors report the detection of a consensus “TIIV-like” amino acid sequence, which induces the assembly of proteins into peroxisomal core structures. The authors further show that mammalian orthologues of *U. maydis* DAO and UOX likewise sorted to luminal subdomains of peroxisomes, when expressed in the fungus, hence, implying a conserved core forming mechanism. Finally, the authors analyzed the functional meaning behind the compartmentalization of peroxisomal enzymes into these different organelle subdomains. Based on the observation that missorting of the core-contained enzymes umMac1 and umMac3 to the matrix disrupts *U. maydis* growth on fatty acids, the authors conclude that peroxisomal subdomains are required to separate enzymes involved in fatty acid β -oxidation from other, more specialized enzymes in order to ensure efficiency of the pathway. In general, the manuscript submitted is written fluently and intelligibly and could be of general interest to a broader scientific community as it proposes a novel, conserved mechanism to separate the organelle proteomes into functionally relevant subdomains. The experiments presented seem to be thoroughly performed and conclusions drawn are per se largely sound. However, when regarded in general, the individual observations and results presented in the manuscript do in my opinion not support the concluding hypothesis of a conserved mechanism for peroxisomal subdomain formation on a grander scale. My concerns are based on the reasons that (1) even in *Ustilago* only a subset of subdomain/core forming proteins possess the TIIV-like motif (the position and amino acid sequence is e.g. for umUox not conserved in other species including other basidiomycetes), the TIIV-motif of Mac3 and the TRIV-motif of Uox1 exhibit significantly different physicochemical properties (hydrophobic vs charged) and that mammalian peroxisomal cores were reported to possess a heterogeneous architecture. In mammals, DAO and UOX do not co-localize in the same peroxisomal cores but assemble in different subdomains inside the peroxisomal matrix. While UOX in rodent liver is exclusively described as the dominating crystalline core constituent of peroxisomes, DAO is in hepatocytes homogeneously distributed inside the peroxisomal matrix, sometimes even showing decreased concentrations in the central core region (Völkl et al. 1988, Angermüller & Fahimi 1988). In kidney proximal tubule cells, which are devoid of the crystalline cores found in hepatocytes, DAO localizes to the central region of peroxisomes in an amorphous aggregate-like structure (Angermüller & Fahimi 1988). Hence, these early reports suggest that core formation for the mammalian UOX and DAO is controlled by two independent mechanisms. In this respect, without providing further experimental evidence, the authors should interpret their data more cautiously and should as well more critically discuss the results which do not support their hypothesis. Accordingly, several additional experimental controls should be performed to further validate and substantiate the authors’ hypothesis. A detailed list with major and minor comments on the manuscript is provided below:

Major comments:

1. Figures 1, 2, 3: A subset of the Mac1/Mac3/Uox1-positive vesicular structures appear to entirely lack an overlap with the

peroxisomal mCherry-SKL signal. To verify that these signals truly arise from inside peroxisomes but not from cytosolic protein aggregations, the authors should as well detect a peroxisomal membrane marker (e.g. PEX14) to evaluate that such signals are inside the peroxisomal matrix. Moreover, another important control to unravel the mechanism of the Mac1/Mac3 subdomain formation would be a peroxisome-import/SKL deficient variant to see if the core-like protein aggregations form also in the cytosol of the *U. maydis* cells.

2. Figure 2a: While GFP-Mac3-positive subdomains are already visible under constitutive expression in Fig.1, the formation of subdomains in Fig. 2a seem to form only after repression of GFP-Mac3 gene expression. To evaluate this discrepancy, the pulse-chase experiment should be repeated monitoring as well time-dependent de novo subdomain formation after a phase of prolonged arabinose repression.

3. Fig. 2f: The authors should provide Pearson correlation coefficients for the localization of GFP-MAC3-V570R compared to WT-GFP-MAC3, as the image provided shows also turquoise punctate GFP-MAC3-V570R-structures, which seem to lack overlap with the mCherry-SKL.

4. Lines 194-196: The observation that GFP-Uox1 assembles in the same subdomains as either Mac1 or Mac3 is an intriguing finding, since all three proteins appear to possess significantly differing intrinsic "targeting" information (Mac3 C-terminal hydrophobic motif, Uox1: positively charged intrinsic tetrapeptide sequence, Mac1: undefined motif). To get further information on the potential impact of different motifs on respective core-forming capacities, it would be helpful to compare if the degree of colocalization differs between the three protein and to matrix marker protein using Pearson correlation.

5. Line 267: the authors state that a TRIV motif in Uox1 would have similar sequence characteristics as the TIIV-motif observed for Mac3. This is in my opinion not justified, since the exchange of an isoleucine by an arginine drastically changes the physicochemical properties of such a short tetrapeptide sequence. Moreover, in contrast to the statement in the manuscript (line 268), the "Aggrescan" predictor used by the authors does not predict the region around the Uox1-TRIV-amino acid motif as prone to aggregation (see graph in the attached document). Additionally, the TRIV motif from umUox1 appears to be only conserved among Ustilaginomycota but it is already absent in other Basidiomycota and animals. In this regard, the molecular mechanism for the subdomain formation remains elusive, as it does not seem to rely on a defined hydrophobic sequence acting as a nucleation center for core formation. In order to confirm the function of the TRIV as a driving motif for peroxisomal subdomain formation, the authors should at least repeat the in vitro translation assay presented in Fig. 2h/j with a HA-tagged GFP-TRIV construct.

6. Lines 287-288: Neither MmUOX nor MmDAO appear to contain TIIV-like sequences. In this light, it is highly surprising that MmUOX and GFP-TIIV-SKL, when expressed in RPE1 cells, colocalize in the same peroxisomal subdomains. Moreover, rodent UOX assembles in a highly ordered crystalline core structure (Völkl et al. 1988.) To confirm that UOX core formation follows an equivalent mechanism, when expressed in *Ustilago*, it would be necessary to confirm its native crystalline structure; therefore, the authors should provide high-resolution immuno-EM images of MmUOX core in *U. maydis*.

7. Fig. S3f: From the images provided it is not clear if the MTS-GFP-TIIV signal is indeed inside mitochondria or localized in protein aggregates adjacent to mitochondria. A mitochondrial outer membrane marker protein might be better suitable to proof mitochondrial import and subdomain formation for the MTS-GFP-TIIV construct.

8. Line 331 ff.: The author state that peroxisomal core formation follows analogous principles in mammals and fungi. To substantiate their hypothesis on a phylogenetic background, murine PTS1-containing proteins should be screened for conserved short hydrophobic peptide sequences comparable to the TIIV-motif from *U. maydis*.

Minor comments.

1. Line 75: "enriched in a fraction of organelles": a better term to describe the heterogeneous protein distribution would be "subset of organelles"

2. Line 79: "incubation of cells in oleic acid enhanced substantially the phenotype above". This sentence is unclear. Do the authors mean the diverging distribution of Mac1 and Mac3 positive peroxisomes compared to m-Cherry-SKL?

3. Line 81: Colors used in the figures are "magenta" and "turquoise" not "red" and "green".

4. Lines 121/122: "A GFP-tagged version of Aox1 was uniformly distributed ... (Fig. S3a)." Actually, the intensities for AOX1 and mCherry-SKL shown in the magnification do not support this conclusion – both channels show intensity maxima at different locations inside the organelle. If this cut-out is not representative for the situation, it should be replaced.

5. Fig. 1f: In order to better illustrate the overlap of the magenta and turquoise in the z-axis of the 3D-reconstructions, the authors should add images, which as well visualize reconstructions in the z-plane.

6. Lines 143/146/148: The figure references should most likely be "Figs. S3c and S3d, S3e, S3f"

7. Line 198: Fig S3a should be Fig S4a.

8. Table S1 and S2 are not in an appropriate, easy-to-read format

9. Table S1: The authors should add columns presenting the actual amino acid sequence of the detected TIIV-like sequences and information on their position in the respective proteins.

Reviewer #3

(Remarks to the Author)

Peroxisomes are small, membrane-bound organelles present in almost all eukaryotic cells. They play a crucial role in cellular metabolism, particularly in the oxidation of fatty acids and the detoxification of hydrogen peroxide. Peroxisomes contain enzymes such as catalase and oxidases, which break down toxic peroxides into water and oxygen. Dysfunction of this organelle can lead to severe metabolic disorders such as Zellweger syndrome, characterized by the accumulation of very long-chain fatty acids and other toxic substances in cells. This results in neurological impairment, liver dysfunction, and developmental abnormalities.

Fungal peroxisomes include specialized metabolic pathways, with *Neurospora crassa*'s Woronin bodies playing a role in

sealing septal pores. In addition to observed peroxisome sub-populations, internal compartmentalization like subdomains of peroxisomal membrane proteins or detergent resistant core structure composed out of urate oxidase or alcohol oxidase, respectively, were reported.

How this internal structure is assembled, remains elusive and is subject of this manuscript. The authors demonstrate the assembly of such core structures, identify conserved sequences in some proteins that are sufficient for core targeting and they assume that the protein assembly to higher cores will increase metabolic effectiveness.

The manuscript and its presented figure are of good quality and the experiments are carried out properly. However, several questions remain unanswered. The paper suffers from the lack of a physiological relevance of the without doubt interesting phenomenon.

Major Comments:

1a) It has been reported that in cellulose crystals of e.g. alcohol oxidase can be generated inside yeast peroxisome just by alteration of the protein concentration (Jakobi et al 2016). Accordingly, the observed core formation could be an uncontrolled process, which might be triggered by mentioned protein motif but has no real physiological relevance. Along, this line, the biosynthesis of surface-active mannosylerythritol lipids MEL production was not affected upon inhibition of core formation and did only result in moderate inhibition of the growth on oleate (not quantified) and O₂ consumption upon oleate consumption. Thus, the physiological relevance of the observed directed aggregation needs to be addressed.

1b) In this context, it also must be confirmed that the introduced mutation does not affect the enzymatic activity of the protein?

2a) The manuscript mostly relies on co-localization of proteins in peroxisomal cores as judged by fluorescence, however, it remains elusive with respect to the nature and formation of the peroxisomal cores. Are the cores formed by homo- or hetero-oligomerization of proteins. Thus, is there an interaction between Mac1 and Mac1 or Mac1 and Mac3? Moreover, Is the formation of the peroxisomal cores simply triggered by aggregation due to a high protein concentration or does the newly identified motif indeed functions as a targeting signal?

2b) What happens in strain that lack peroxisomes? If the identified motif is an interaction site, one would expect that condensation would also take place in the cytosol? What happens when either one of these proteins is absent (deletion strain).

3) The very short discussion remains superficial; it is just a repetition of the results. A discussion of the possible physiological relevance of the observed core formation is missing.

4) In the discussion, it is mentioned that the mutated variants of Uox1 and Mac3 strongly affect cell metabolism, which is assumed to be due to lack of condensation. Are these variants still biological active? The introduced amino-acid change might blocks the enzyme activity, which

Minor Comments:

Figure 1a vs b: The growth conditions should be defined? What is the promotor in the constructs use in Figure 1? Like in figure 2 first growth on arabinose then shifted to oleate or glucose? At which time-points the images were taken?

Figure 2: The TIIV-motif of Mac3 might trigger aggregation. Mac1 lacks this motif but is also locally concentrated. How do the authors explain this finding? The authors mention an embedding of the information within Mac1 (line 140). Is it also likely that Mac1 and 3 interact with each other and only one contains concentration information?

Figure 2/ line 121: Is the acyl-CoA construct under control of the same promotor (arabinose on, glucose off)? This is not clear from the text.

Figure 4f/5d: Immunoblots should always be shown with two Mw-markers.

Lines 681/2: Reference is incomplete.

Lines 696/7: Reference is incomplete.

Lines 699/700: Reference is incomplete.

Lines 768/9: Reference is incomplete.

Supplementary Figure S3a: scale bar is missing.

Reviewer #4

(Remarks to the Author)

Version 1:

Reviewer comments:

Reviewer #1

(Remarks to the Author)

The authors have satisfactorily addressed the issues I raised previously.

Reviewer #2

(Remarks to the Author)

In my opinion the additional control experiments performed for the revised version of the manuscript "Peroxisomal core structures segregate diverse metabolic pathways" submitted by Freitag and colleagues significantly improved the scientific robustness of the work. Moreover, the extension of the discussion now much more critically evaluates its potential and limitations with respect to its scientific impact for the research field. In conclusion, I would recommend the acceptance of the manuscript in its present form.

Reviewer #3

(Remarks to the Author)

In the revised manuscript, the authors tried to address all concerns of this reviewer.

It is demonstrated that mutated proteins exhibit physiological enzyme activity but fail to complement mutant strain most likely due to lack of organization into core structures. These structures are initiated in case of Mac3 by a newly identified protein motive.

The main concern addresses the physiological relevance of the finding, which is only slightly addressed. Why are mutated proteins functional (enzymatically active) but fail to be active in vivo just due to lack of condensation? The authors themselves state at several points of their rebuttal letter that "the understanding of the physiological relevance of cores is only at the beginning". Moreover, a general rule of condensate formation outside *Ustilago* is not supported. The authors mention, that "it is too early to definitely conclude that core formation is used for metabolic compartmentalization of peroxisomes in many species".

At the end, I have the feeling that, despite the well-executed and documented experiments, without a clear functional relevance it is too early to derive a fundamental principle from this work. This makes it difficult to recognize the general interest for a broader readership even. Thus, at this stage the work is suited for a more specialised journal.

Reviewer #4

(Remarks to the Author)

First, we would like to thank all reviewers for their constructive comments, which we could mostly address.

1. We now demonstrate enzymatic activity of non-core Uox1 (suggested by reviewer 3).
2. We demonstrate that purified non-core Uox1 behaves like non-core Mac3 again speaking for self-assembly as a potential mechanism for core formation (derived from suggestions of all reviewers).
3. We show that Mac3 is less stable compared to Aox1 but that this effect appears to be independent of the core phenotype (suggested by reviewer 1).
4. We performed more imaging experiments and quantifications to further confirm our microscopic observations (suggested by reviewers 1,2 and 4).
5. We show now that accumulation of Mac1 and Mac3 is independent of each other (suggested by reviewer 3).
6. We show that cores form less efficient in the cytosol (suggested by reviewers 2, 3 and 4).
7. We made the textual and formal edits as requested.

This significantly improved our paper.

In this manuscript, Bäcker and colleagues looked at how and why peroxisomes may compartmentalize their protein content. The hypothesis is that the “sub-compartments” observed in peroxisomes from some organisms may provide a functional advantage by segregating metabolic pathways. This is an important subject and the work described in this manuscript has great potential to advance our knowledge on peroxisome biology. However, while the author's hypothesis is plausible, the data supporting it needs to be improved. The authors should address the following points:

Major issues:

1- Lines 34-36, 59-64, 233-239, and 351-353 – The idea that peroxisomes from many species contain well defined functional sub-domains in their matrix should be toned down. Note that crystalline cores of UOX in rat liver contain essentially UOX and no other enzymes (e.g., PMID: 1594592). In agreement with this, ref. 35 shows that, except for UOX, all the other enzymes studied are soluble in the presence of detergent. Also, crystals of alcohol oxidase (AOX) from Pichia pastoris/Hansenula polymorpha can form in vivo when expressed at high levels, even in the absence of peroxisomes, suggesting that this is an intrinsic property of the protein (PMID: 27006771; PMID: 27458710 and references cited therein). Finally, the enzymatic content of rodent and human peroxisomes is almost identical with the exception that humans lack UOX (and peroxisomal crystalloids). If the function of these crystalloids were to compartmentalize non-beta-oxidation enzymes, humans would be in trouble.

We agree with the reviewer that it is too early to definitely conclude that core formation is used for metabolic compartmentalization of peroxisomes in many species and thus toned down this claim. To finally prove the idea we would need e.g. mutated versions of murine Uox1 or human DAO not enriched in cores anymore (just as the versions we obtained for *U. maydis* Uox1 and Mac3), and we feel that this is outside the scope of this paper but something definitely worth to address. Nevertheless, proteins we have tested that have previously been observed in cores (e.g. murine Uox1) also co-localized with *U. maydis*. There appears additional conservation of core resident enzymes e.g. D-amino acid oxidase in fungi and in mammals. Motifs triggering core formation are probably more diverse (Fig. 5h). We do not think that all these proteins make crystalloids, however, they all are able to self-assemble in detergent resistant aggregates. Hence, many peroxisomal proteins may share the ability to self-assemble.

2- Lines 117-118 – Conceptually, intraperoxisomal selective degradation should also be considered here. In the pulse chase experiment shown in Fig 2a there is an apparent decrease in the fluorescence of GFP-Mac3 over time. Is it possible that soluble Mac3 is constitutively degraded in peroxisomes while the aggregated Mac3 pool resists proteolysis? The authors should check Mac3 protein level over time by western blotting.

Also, in Fig. S3b, at t=3h, almost all magenta and cyan dots overlap, yet at 6h the authors show a cell in which most peroxisomes have mCherry-SKL but lack GFP-Mac3. According to the explanation provided by the authors (lines 130-131), this would imply that peroxisomes more than doubled their numbers in 3 hours. But if this is true, then the fluorescence intensity of GFP-AOX1 in relation to mCherry-SKL should be decreased by a factor >2. This decrease cannot be perceived in fig. S3b. Did the authors check this? Otherwise, the different subpopulations of peroxisomes may be better explained by degradation of Mac3, and not by the unequal distribution of the aggregates during division.

This is an interesting point and we thank the referee for the notion. Following up on this advanced our understanding of Mac3 properties. Peroxisomes probably more than doubled in the three hours as the fungus usually divides within two hours. Nevertheless, it needs some time until the mRNA is totally gone explaining the microscopy and immunoblot data for GFP-Aox1. We find that GFP-Mac3 is less stable compared to GFP-Aox1 (Fig. 2c). A peroxisomal protease we are aware of is a putative LON protease. Deletion of the corresponding gene did not affect segregation or core formation (Fig. S4) – thus destabilization is independent of core formation or independent of LON. Although we consider destabilization of Mac3 as an interesting finding it seems not relevant for the phenotype of strains expressing more homogeneously distributed variants of Mac3 and Uox1, respectively. We observe similar steady-state protein levels (Figs. 4f and 5e). If it was only the aggregated versions of Mac3 or Uox1, sustaining proteolysis, we would expect significant destabilization of the variants distributed in the entire lumen. Taken together, although this finding appears not directly relevant to our core phenotype at a first glance we think it is very interesting and show the data in a main figure. Mac3 might again prove useful to understand selective degradation of peroxisomal matrix proteins – a process far from being well-characterized. It would make sense that proteins that need to segregate to enable fatty acid breakdown are less stable as well.

3- Data in Fig. S3c - Both GFP-9mer-SKL and GFP-7mer-SKL already display puncta that do not localize precisely with mCherry-SKL. It is unclear whether these non-localizing puncta are independent organelles or subdomains of the same organelle. Please clarify. More strikingly, there is also heterogeneity in mitochondria containing MTS-GFP and MTS-mCherry (Fig. S3f) – in the “overlap” inset of that figure, one can see zones with “pure” cyan and “pure” red. The heterogeneity in this negative control is difficult to understand, when such differences are sometimes interpreted as segregation (e.g. Fig. 6 and c). Please clarify.

Please note that the pictures depicted in Fig. S5 were obtained by epifluorescence microscopy, while e.g. Fig. 6c is SIM data. For epifluorescence microscopy we sometimes observe slight shifts in the channels, however, the quantifications of Pearson correlation coefficients and also the overall appearance of the pictures is enough to identify the core phenotype. We now also provide SIM for the mitochondria experiment to show that foci tightly associate with mitochondria and to highlight the striking difference with a normal Mito marker protein (Fig. S5d). More generally, we included SIM imaging for several key findings or where required, and e.g. for chase experiments show data acquired by both approaches to highlight that the results merge in principle.

4- It is surprising that cells expressing either GFP-Mac3-V570R or GFP-Uox1-V124R display the same phenotype. While one can see the link between the acyltransferase and β -oxidation, it is harder to see how Uox1-V124R would affect growth in oleic acid, especially considering that other “core-forming” peroxisomal proteins should still form these structures. Is the idea that any one abundant core-forming protein will seed the aggregation/crystallization of others? The authors should discuss their model in more detail.

Given that mixed structures that form upon co-expression it could well be that the core-forming proteins affect each other to some extent. A notable but so far unexplained exception is Mls1 (Figs. 5i and S10d). Nevertheless, it could also be that it is each delocalized enzyme alone, which perturbs peroxisomal fatty acid metabolism. Please note that also the mutated (more soluble) Uox1 variant is enzymatically functional (Fig. 5f). In addition, Uox1 variants behave very similar to Mac3 variants in aggregation assays (Figs. 2i and 5b). We are currently studying, how further enzymes occurring in cores (Fig. 5h) can be distributed more regularly inside of the peroxisomal lumen e.g. by mutation of surface resident TIIV-like motifs or by other mutations and how this affects peroxisomal metabolism. This hopefully develops into a follow-up paper, which may reveal if the herein proposed β -oxidation compartment needs to be cleared from additional metabolic enzymes (Fig. 7; Table S2) and how different core forming proteins behave upon mixed challenges (e.g. urate and fatty acids). We expanded our discussion section to emphasize that our paper probably is door opener to further explore metabolic compartmentalization of peroxisomes in *U. maydis*. But we also discuss what we still don't know. Do these cores preferentially evolve to form (sometimes even as crystals) if co-existence of distinct biochemical pathways is required regularly? This is intuitive for Mac enzymes, which depend on β -oxidation (PMID: 38377076) or urate oxidase in peroxisomes of the liver, which catabolizes fatty acids and urate simultaneously.

Other issues:

5- Lines 47-48 – “through a hydrogel-like channel resembling the interior of the nuclear pore complex”.

This is not a fact. This is simply one of many very hypothetical models presently available on how proteins translocate the peroxisomal membrane (e.g., PMID: 38936257).

We agree and introduce this as a possibility and not as a fact.

6- Section "A short peptide motif in Mac3..." - please note that alphafold predicts with high confidence the structure of Mac3 (<https://alphafold.ebi.ac.uk/entry/A0A0D1EB13>). In the predicted model the TIIV motif is part of an alpha-helix and the 3 hydrophobic residues of the motif (Ile-Ile-Val) face the hydrophobic core of the protein, as expected. Thus, it is very difficult to understand how such a motif might work. Some speculations in the discussion would be welcome.

We show in aggregation assays with purified proteins that mutation of the motifs in both, Mac3 and Uox1, greatly reduces their ability to aggregate. Self-assembly, therefore, seems a reasonable explanation. It cannot only be the hydrophobicity of these motifs as the threonine residue at position 1 is important (Fig. S6a and b). We expanded the discussion section to clarify that identification of TIIV motif enabled us to get access to the phenomenon of metabolic compartmentalization, but we still miss mechanistic details. We will certainly be able to provide more information when we have more data on further candidates identified in cores (Fig. 5h).

7- Lines 208-209 – the authors have not shown that U. maydis GFP-Uox1 forms paracrystalline structures. Thus, the conclusion in this last sentence is not supported by the data. Please rephrase.

This has been changed to be more precise. We wanted to point out that it appears to be a similar process.

8- Lines 96-98 – Please provide a short explanation of why the GFP-mCherry-SKL proteins appear as rings in the 3D-reconstruction (Fig. 1f) for those not familiar with these techniques.

This information has been added in the Methods section.

9- Fig S3f - the mitochondrial localization of MTS-GFP-TIIV should be supported with more robust data (e.g., using a mitochondrial outer membrane marker). As presented, there may be doubts on the localization of this protein (mitochondrial or cytosolic?)

We performed SIM for this issue and see that the protein clearly co-localized with mitochondria. If all foci represent luminal protein or if some aggregates also stick to the exterior of these organelles is not entirely clear but also not super relevant to this paper. We think that for formation of cores the local concentration matters – we only observe inefficient accumulation upon cytosolic overexpression but efficient accumulation upon mitochondrial targeting (Fig. S5d).

*10- Some figures are misidentified in the main text:
- Fig.S3 c to i are misidentified as Fig.S2 at several places
- In line 198, S3a should be S4a
- In line 204 S3b should be S4b
- In line 258, e should be f
- Figure S7 and S6 are swapped.*

We apologize for these mistakes. They have been addressed.

Reviewer #2 (Remarks to the Author):

Dear editors,

*the manuscript „Peroxisomal core structures segregate diverse metabolic pathways” submitted by Freitag and colleagues reports the discovery of a phylogenetically conserved mechanism separating peroxisomal matrix proteins into homogeneously distributed enzymes or centrally concentrated core structures. The results presented are primarily based on experiments performed in the filamentous fungi *Ustilago maydis*, in which the authors report the detection of a consensus "TIIV-like" amino acid sequence, which induces the assembly of proteins into peroxisomal core structures. The authors further show that mammalian orthologues of *U. maydis* DAO and UOX likewise sorted to luminal subdomains of peroxisomes, when expressed in the fungus, hence, implying a conserved core forming mechanism. Finally, the authors analyzed the functional meaning behind the*

compartmentalization of peroxisomal enzymes into these different organelle subdomains. Based on the observation that missorting of the core-contained enzymes umMac1 and umMac3 to the matrix disrupts U. maydis growth on fatty acids, the authors conclude that peroxisomal subdomains are required to separate enzymes involved in fatty acid β -oxidation from other, more specialized enzymes in order to ensure efficiency of the pathway. In general, the manuscript submitted is written fluently and intelligibly and could be of general interest to a broader scientific community as it proposes a novel, conserved mechanism to separate the organelle proteomes into functionally relevant subdomains. The experiments presented seem to be thoroughly performed and conclusions drawn are per se largely sound. However, when regarded in general, the individual observations and results presented in the manuscript do in my opinion not support the concluding hypothesis of a conserved mechanism for peroxisomal subdomain formation on a grander scale. My concerns are based on the reasons that (1) even in Ustilago only a subset of subdomain/core forming proteins possess the TIIV-like motif (the position and amino acid sequence is e.g. for umUox not conserved in other species including other basidiomycetes), the TIIV-motif of Mac3 and the TRIV-motif of Uox1 exhibit significantly different physicochemical properties (hydrophobic vs charged) and that mammalian peroxisomal cores were reported to possess a heterogeneous architecture. In mammals, DAO and UOX do not co-localize in the same peroxisomal cores but assemble in different subdomains inside the peroxisomal matrix. While UOX in rodent liver is exclusively described as the dominating crystalline core constituent of peroxisomes, DAO is in hepatocytes homogeneously distributed inside the peroxisomal matrix, sometimes even showing decreased concentrations in the central core region (Völkl et al. 1988, Angermüller & Fahimi 1988). In kidney proximal tubule cells, which are devoid of the crystalline cores found in hepatocytes, DAO localizes to the central region of peroxisomes in an amorphous aggregate-like structure (Angermüller & Fahimi 1988). Hence, these early reports suggest that core formation for the mammalian UOX and DAO is controlled by two independent mechanisms. In this respect, without providing further experimental evidence, the authors should interpret their data more cautiously and should as well more critically discuss the results which do not support their hypothesis.

We appreciate this criticism and the careful thoughts of both reviewers and discuss the data more appropriately in the revised paper to better clarify what we know and what we don't know. We did not intend to define TIIV-like motifs as a "consensus" for subdomain targeting. They are simply the first examples of elements necessary and sufficient for this phenomenon to happen. Given the simple nature of our motif redundancy on the level of the sequence is likely and already corroborated by our data (Fig. S6 and Fig. 5h). We tried to point out this in the initially submitted manuscript: "While these *in vitro* results apparently align well with our *in vivo* observations, the TIIV motif may be part of a more complex mechanism, warranting further studies. Nonetheless, our study identifies TIIV as a key motif driving the self-assembly of a protein and compartmentalization in peroxisomal foci." We extended the discussion in the updated manuscript. Position probably matters and TIIV-like motifs can be false positives (Fig. 5h). Nevertheless, the discovery of functional motifs significantly helped to understand what is going on: our study showcases how small perturbations can affect core formation and oleic acid metabolism, albeit they leave the enzymatic function intact.

In the remainder of the study we find that there are proteins with overlapping functions (with or without TIIV) that tend to accumulate in cores in fungi and in mammals. Key examples are D-amino acid oxidase and urate oxidase, which are probably in peroxisomes because they make H_2O_2 , but are not directly related to fatty acid breakdown. We think that these parallels and the wealth of functionally different candidates we identified are good hints to suggest that cores are enriched for many enzymes facilitating more auxiliary functions of peroxisomes, which can be more specific to certain group of organisms (Woronin body protein HexA, methanol oxidase, MEL biosynthesis enzymes) or conserved (urate oxidase, d-amino-oxidase).

Accordingly, several additional experimental controls should be performed to further validate and substantiate the authors' hypothesis. A detailed list with major and minor comments on the manuscript is provided below:

Major comments:

1. Figures 1, 2, 3: A subset of the Mac1/Mac3/Uox1-positive vesicular structures appear to entirely lack an overlap with the peroxisomal mCherry-SKL signal. To verify that these signals truly arise from inside peroxisomes but not from cytosolic protein aggregations, the authors should as well detect a peroxisomal membrane marker (e.g. PEX14) to evaluate that such signals are inside the peroxisomal matrix. Moreover, another important control to unravel the mechanism of the Mac1/Mac3 subdomain

formation would be a peroxisome-import/SKL deficient variant to see if the core-like protein aggregations form also in the cytosol of the U. maydis cells.

Please note that we sometimes observe shifts between the red and the green signal in epifluorescence imaging – this does not change the assessment of the overall phenotype (core or not). We performed SIM imaging for several cases and also include Pex12 as an additional marker (Fig. S2e and f). In aggregate our data clearly shows that most foci are inside peroxisomes and that they start to form inside of the lumen is confirmed by the chase experiment (Fig. 2a). In the cytosol, core formation is less efficient (Fig. S5f). We therefore assume that self-assembly occurs more efficient after import upon concentration in the limited volume of peroxisomes or even mitochondria. From some of our SIM images we speculate that the signal may sometimes be at the outside, which together with the chase data would leave the possibility that cores can be exported or budded of. We need to follow up on this as such a mechanism would be a surprise. They might as well only be in the periphery but still inside.

2. Figure 2a: While GFP-Mac3-positive subdomains are already visible under constitutive expression in Fig. 1, the formation of subdomains in Fig. 2a seem to form only after repression of GFP-Mac3 gene expression. To evaluate this discrepancy, the pulse-chase experiment should be repeated monitoring as well time-dependent de novo subdomain formation after a phase of prolonged arabinose repression.

We are not sure if we understand this point correctly. Our chase experiment shows the *de novo* formation of subdomains/cores after glucose repression leading to a pulse of arabinose induction. First, newly made GFP-Mac3 is uniformly distributed in peroxisomes and becomes sequestered into subdomains over time. In our view, this is in agreement with the constitutive expression experiment as there is always older protein in peroxisomes already accumulated in cores. We performed the experiment to decide if cores form before or after import. According to this data it occurs after import.

3. Fig. 2f: The authors should provide Pearson correlation coefficients for the localization of GFP-MAC3-V570R compared to WT-GFP-MAC3, as the image provided shows also turquoise punctate GFP-MAC3-V570R-structures, which seem to lack overlap with the mCherry-SKL.

This has been added and confirms a significant difference between both protein variants with respect to core formation (Figs. 2g and S6d). In addition, we show that GFP-Mac3-V570R and mCherry-Mac3 segregate (Fig. 2g).

4. Lines 194-196: The observation that GFP-Uox1 assembles in the same subdomains as either Mac1 or Mac3 is an intriguing finding, since all three proteins appear to possess significantly differing intrinsic “targeting” information (Mac3 C-terminal hydrophobic motif, Uox1: positively charged intrinsic tetrapeptide sequence, Mac1: undefined motif). To get further information on the potential impact of different motifs on respective core-forming capacities, it would be helpful to compare if the degree of colocalization differs between the three protein and to matrix marker protein using Pearsson correlation.

Our quantification data on the different candidates show that the ability to cluster is very similar (e.g. Fig. 1c and Figs. 5a and 5h). However, before knowing the other motifs/triggers e.g. in Mac1 the reasons for that are hard to predict. We measured co-localization for several pairs after preparation of crude organelles and Triton X-100 (Mac3 – Mac1; Fig. S7c and Mac1 – Uox1; Fig. S7a). They overlap pretty well. The only exception we found so far is malate synthase Mls1 which is accumulating in cores but does not entirely overlap with the other candidates tested for co-localization. This resembles the phenomenon described by Fahimi and colleagues which we discuss in the updated manuscript. In addition, cores containing Uox1 and Mac proteins may rarely form under physiological conditions as glycolipid biosynthesis occurs upon nitrogen deprivation and uric acid is a nitrogen source. The same might be true for Dao1. In contrast all these processes are likely to co-occur with fatty acid breakdown or glyoxylate metabolism, which gives a hint on the functional relevance of the different modes of segregation we observe.

5. Line 267: the authors state that a TRIV motif in Uox1 would have similar sequence characteristics as the TIIV-motif observed for Mac3. This is in my opinion not justified, since the exchange of an isoleucine by an arginine drastically changes the physicochemical properties of such a short tetrapeptide sequence. Moreover, in contrast to the statement in the manuscript (line 268), the “Aggrescan” predictor used by the authors does not predict the region around the Uox1-TRIV-amino

acid motif as prone to aggregation (see graph in the attached document). Additionally, the TRIV motif from umUox1 appears to be only conserved among Ustilaginomycota but it is already absent in other Basidiomycota and animals. In this regard, the molecular mechanism for the subdomain formation remains elusive, as it does not seem to rely on a defined hydrophobic sequence acting as a nucleation center for core formation. In order to confirm the function of the TRIV as a driving motif for peroxisomal subdomain formation, the authors should at least repeat the in vitro translation assay presented in Fig. 2h/j with a HA-tagged GFP-TRIV construct.

We apologize for this mistake and thank the reviewers for the notion. Indeed, TRIV it is not in a predicted aggregation prone region according to Aggrescan, we just looked in the wrong line. Hence, we were lucky with the results. We performed aggregation assays for purified GFP-TRIV, for Uox1 and the respective non-core variant Uox1-V₁₂₄R. We obtained results similar to our data on purified Mac3 variants or GFP-TIIV (Figs. 2i and j and 5b). Hence, our data became stronger also on the mechanistic side as changes of the motifs have similar effects *in vivo* and *in vitro* for two very distinct enzymes.

6. Lines 287-288: Neither MmUOX nor MmDAO appear to contain TIIV-like sequences. In this light, it is highly surprising that MmUOX and GFP-TIIV-SKL, when expressed in RPE1 cells, colocalize in the same peroxisomal subdomains. Moreover, rodent UOX assembles in a highly ordered crystalline core structure (Völkl et al. 1988.) To confirm that UOX core formation follows an equivalent mechanism, when expressed in Ustilago, it would be necessary to confirm its native crystalline structure; therefore, the authors should provide high-resolution immuno-EM images of MmUOX core in U. maydis.

There are redundant signals for core-assembly (Fig. 5h). TIIV in Mac3 or TIRV in Uox1 are examples necessary and sufficient for this process to occur. Identification of such motifs was very valuable to functionally address formation of the subdomains and to learn that assembly via these motifs is likely to facilitate metabolic compartmentalization inside of the peroxisomal lumen. Please note that we do not know or suggest that all of these proteins are able to form crystals. However, proteins known to form crystals clearly localize in subdomains upon expression in *U. maydis* (e.g. murine Uox1 or fungal HexA). We expanded the results and discussion to emphasize that the ability to accumulate in a detergent resistant core and the ability to build the crystalline structure are different characteristics, which might well support each other. For yet unknown reasons we do not obtain samples with preserved ultrastructure for strains expressing GFP-tagged murine Uox1 in *U. maydis*. However, we are confident that our data concerning the overlaps in the different biological systems are already a good hint to suggest that segregation occurs for enzymes with overlapping functions in *U. maydis* and in human cells (Uox and D-amino acid oxidase, respectively). We are going to work on the phenotype of murine cells expressing Uox1 variants, which fail to form cores, but this will lay the grounds for a new paper.

7. Fig. S3f: From the images provided it is not clear if the MTS-GFP-TIIV signal is indeed inside mitochondria or localized in protein aggregates adjacent to mitochondria. A mitochondrial outer membrane marker protein might be better suitable to proof mitochondrial import and subdomain formation for the MTS-GFP-TIIV construct.

We performed SIM to tackle this issue. It accumulates very close to the mitochondria (Fig. S5d). Together with our data on cytosolic variants of TIIV containing proteins (Fig. S5f) we conclude that local concentration matters for accumulation in cores. We cannot exclude that some MTS-GFP-TIIV sticks to the outside of mitochondria, but we think that this is not relevant to make our point of self-assembly upon high local concentration. The reduction of complexity from 3D (free diffusion in the cytosol) to 2D (targeting to the mitochondrial membrane) might be enough.

8. Line 331 ff.: The author state that peroxisomal core formation follows analogous principles in mammals and fungi. To substantiate their hypothesis on a phylogenetic background, murine PTS1-containing proteins should be screened for conserved short hydrophobic peptide sequences comparable to the TIIV-motif from U. maydis.

We did this mining and indeed find that similar enzymes contain such motifs e.g. a protein with similarity to epoxide hydrolases (Table S2). However, given that in *U. maydis* many proteins without a TIIV like motif cluster in cores, we expect that our findings only reflect the tip of an iceberg and there is still much to learn about peroxisomal compartmentalization and the formation of peroxisome subpopulations.

Minor comments.

1. Line 75: “enriched in a fraction of organelles”: a better term to describe the heterogeneous protein distribution would be “subset of organelles”

Thanks. We changed this accordingly.

2. Line 79: “incubation of cells in oleic acid enhanced substantially the phenotype above”. This sentence is unclear. Do the authors mean the diverging distribution of Mac1 and Mac3 positive peroxisomes compared to m-Cherry-SKL?

You are right. The sentence was changed.

3. Line 81: Colors used in the figures are “magenta” and “turquoise” not “red” and “green”.

Thanks for spotting this mistake. This was corrected.

4. Lines 121/122: “A GFP-tagged version of Aox1 was uniformly distributed ... (Fig. S3a).” Actually, the intensities for AOX1 and mCherry-SKL shown in the magnification do not support this conclusion – both channels show intensity maxima at different locations inside the organelle. If this cut-out is not representative for the situation, it should be replaced.

Please note that this is epifluorescence data, but it is clear that the localization is very different from e.g. Mac3 (Fig. 1). We rephrased in regularly distributed among mCherry-SKL positive peroxisomes. That it is more or less uniformly distributed is obvious in the SIM data obtained in the course of a similar chase experiment (Fig. 2a).

5. Fig. 1f: In order to better illustrate the overlap of the magenta and turquoise in the z-axis of the 3D-reconstructions, the authors should add images, which as well visualize reconstructions in the z-plane.

This has been added to a supplementary Figure (Fig. S2g).

6. Lines 143/146/148: The figure references should most likely be “Figs. S3c and S3d, S3e, S3f”

7. Line 198: Fig S3a should be Fig S4a.

8. Table S1 and S2 are not in an appropriate, easy-to-read format

9. Table S1: The authors should add columns presenting the actual amino acid sequence of the detected TIV-like sequences and information on their position in the respective proteins.

Again, we thank the reviewer for spotting our mistakes. All other changes requested were introduced into the revised paper.

Reviewer #3 (Remarks to the Author):

Peroxisomes are small, membrane-bound organelles present in almost all eukaryotic cells. They play a crucial role in cellular metabolism, particularly in the oxidation of fatty acids and the detoxification of hydrogen peroxide. Peroxisomes contain enzymes such as catalase and oxidases, which break down toxic peroxides into water and oxygen. Dysfunction of this organelle can lead to severe metabolic disorders such as Zellweger syndrome, characterized by the accumulation of very long-chain fatty acids and other toxic substances in cells. This results in neurological impairment, liver dysfunction, and developmental abnormalities.

Fungal peroxisomes include specialized metabolic pathways, with Neurospora crassa's Woronin bodies playing a role in sealing septal pores. In addition to observed peroxisome sub-populations, internal compartmentalization like subdomains of peroxisomal membrane proteins or detergent resistant core structure composed out of urate oxidase or alcohol oxidase, respectively, were reported. How this internal structure is assembled, remains elusive and is subject of this manuscript. The authors demonstrate the assembly of such core structures, identify conserved sequences in some proteins that are sufficient for core targeting and they assume that the protein assembly to higher cores will increase metabolic effectiveness.

The manuscript and its presented figure are of good quality and the experiments are carried out properly. However, several questions remain unanswered. The paper suffers from the lack of a physiological relevance of the without doubt interesting phenomenon.

Major Comments:

1a) *It has been reported that in cellulose crystals of e.g. alcohol oxidase can be generated inside yeast peroxisome just by alteration of the protein concentration (Jakobi et al 2016). Accordingly, the observed core formation could be an uncontrolled process, which might be triggered by mentioned protein motif but has no real physiological relevance. Along, this line, the biosynthesis of surface-active mannosylerythritol lipids MEL production was not affected upon inhibition of core formation and did only result in moderate inhibition of the growth on oleate (not quantified) and O₂ consumption upon oleate consumption. Thus, the physiological relevance of the observed directed aggregation needs to be addressed.*

We agree with the reviewer that our understanding of the physiological relevance of cores is only at the beginning. However, we are, positive that our data demonstrating that delocalized Mac3 as well as delocalized Uox1 perturb peroxisomal function are a strong starting point for a more detailed understanding of a topic in peroxisome research, which is at the beginning. Please note that *U. maydis* cells are not able to ferment and we don't see a phenotype on glucose but specifically on oleic acid medium. We did not measure OD as this is obscured by the stress filaments produced by the mutants. Both Mac3 variants and both Uox1 variants occur in similar amounts, but only the mutants unable to form cores show the phenotype we describe (Figs. 4d-g and 5b-g). To our knowledge this is the first demonstration that organization of peroxisomal proteins into core structures can serve to support peroxisomal metabolism. Based on this data it should be possible to systematically dissect how the interior of peroxisomes is organized for efficient metabolism e.g. by generation of similar core assembly mutants retaining enzymatic activity (Figs. 4 and 5).

1b) *In this context, it also must be confirmed that the introduced mutation does not affect the enzymatic activity of the protein?*

For GFP-Mac3-T₅₇₀R we confirmed the function *in vivo* as it enables production of almost identical MELs if compared to GFP-Mac3 (Fig. 4a and b). In addition, we purified His-tagged versions of Uox1 and Uox1-TRIR during revision of the paper and confirmed that both proteins are functional *in vitro* and behave very similar to Mac3 variants in aggregation assays (Figs. 5b and f). Both results support our model that it is not the absence of enzymatic activity but disturbance of the β -oxidation pathway triggered by localization in the wrong sub-compartment.

2a) *The manuscript mostly relies on co-localization of proteins in peroxisomal cores as judged by fluorescence, however, it remains elusive with respect to the nature and formation of the peroxisomal cores. Are the cores formed by homo- or hetero-oligomerization of proteins. Thus, is there an interaction between Mac1 and Mac1 or Mac1 and Mac3? Moreover, Is the formation of the peroxisomal cores simply triggered by aggregation due to a high protein concentration or does the newly identified motif indeed functions as a targeting signal?*

The referee is correct that most of the data stems from *in vivo* experiments, but we already had some valuable information on the mechanism of assembly and obtained novel data during revision of the paper. Our data suggest that cores form upon elevated local concentration via self-assembly (chase experiment). Furthermore, we show that the motif we identified as relevant for focal accumulation of GFP-Mac3 induces the ability of Mac3 to form aggregates *in vitro* (Fig. 2i and j). We also demonstrate this for Uox1 in the revised version of the paper corroborating our model of self-assembly supported by the small signals we discovered (Fig. 5b).

2b) *What happens in strain that lack peroxisomes? If the identified motif is an interaction site, one would expect that condensation would also take place in the cytosol? What happens when either one of these proteins is absent (deletion strain).*

We thank the reviewer for this interesting comment and addressed the points. Aggregation can also happen in the cytosol albeit to a much lower extent (Fig. S5f). Thus, high local concentration (certainly found in the organelle lumen) may induce core formation. Furthermore, we prepared organelles from a strain expressing mCherry-Mac3 in $\Delta mac1$ cells and were still able to detect detergent resistant mCherry-Mac3 (Fig. S7d, middle panel). The same was true for GFP-Mac1 in $\Delta mac3$ cells (Fig. S7d, right panel). Thus, it is not an interaction between these proteins triggering accumulation of Mac1 in cores. It is more likely that Mac1 can also self-assemble – this is in line with the finding that many proteins accumulating in cores do not contain a TIIV-like motif (e.g. Fig. 5h)

3) *The very short discussion remains superficial; it is just a repetition of the results. A discussion of the possible physiological relevance of the observed core formation is missing.*

We agree with the reviewer and significantly expanded the discussion section.

We address the following topics:

1. The role of this peroxisomal compartmentalization and its potential functional conservation.
2. The structure of the cores and the requirement of TIIV like motifs.
3. An outlook on subpopulations of organelles in general.

4) *In the discussion, it is mentioned that the mutated variants of Uox1 and Mac3 strongly affect cell metabolism, which is assumed to be due to lack of condensation. Are these variants still biological active? The introduced amino-acid change might blocks the enzyme activity, which*

The activity remained unchanged as far as we can tell. Please refer to comment 1b.

Minor Comments:

Figure 1a vs b: The growth conditions should be defined? What is the promotor in the constructs use in Figure 1? Like in figure 2 first growth on arabinose then shifted to oleate or glucose? At which time-points the images were taken?

We defined the growth conditions in the Mat Met section. Please note that all constructs are expressed under control of the constitutive *otef*-promoter if not mentioned otherwise. To prove that it is a natural occurring phenomenon we tested core formation of GFP-Mac1 under control of its natural promoter (Fig. 1d). This promoter is induced upon nitrogen depletion (PMID: 15932999).

Figure 2: The TIIV-motif of Mac3 might trigger aggregation. Mac1 lacks this motif but is also locally concentrated. How do the authors explain this finding? The authors mention an embedding of the information within Mac1 (line 140). Is it also likely that Mac1 and 3 interact with each other and only one contains concentration information?

Please refer to point 2b.

Figure 2/ line 121: Is the acyl-CoA construct under control of the same promotor (arabinose on, glucose off)? This is not clear from the text.

Thanks for this point, we have modified the text to make this clear.

Figure 4f/5d: Immunoblots should always be shown with two Mw-markers.

Lines 681/2: Reference is incomplete.

Lines 696/7: Reference is incomplete.

Lines 699/700: Reference is incomplete.

Lines 768/9: Reference is incomplete.

Supplementary Figure S3a: scale bar is missing.

We apologize for these mistakes – all these points were addressed.